# GUI-SHIFT: ENHANCING VLM-BASED GUI AGENTS THROUGH SELF-SUPERVISED REINFORCEMENT LEARNING

**Longxi Gao[1], Li Zhang[1], Pengzhi Gao[2], Wei Liu[2], Jian Luan[2], Mengwei Xu[1]***

[1]Beijing University of Posts and Telecommunications [2]Independent Researcher
glx@bupt.edu.cn

## ABSTRACT

Training effective Vision-Language Models (VLMs) for GUI agents typically depends on large-scale annotated datasets, whose collection is both labor-intensive and error-prone. We introduce $K$**-step GUI Transition**, a self-supervised inverse dynamics task in which VLMs learn GUI dynamics by predicting the initial action that causes a transition between two GUI states. This approach eliminates the need for natural language instructions and enables scalable dataset construction from existing GUI trajectories or automated exploration. Building on this task, we propose **GUI-Shift**, a reinforcement learning (RL) framework that combines rule-based optimization with data filtering to improve VLM performance. We conduct extensive experiments using multiple VLM backbones across five benchmarks, spanning GUI task automation (AndroidControl, GUI Odyssey, AndroidWorld) and GUI grounding (ScreenSpot-v2, ScreenSpot-Pro). Our results show that training on GUI-Shift generalizes well to both GUI automation and grounding tasks, yielding up to an 11.2% increase in GUI automation accuracy. This study underscores the potential of self-supervised RL to leverage unlabeled GUI trajectories and offers a scalable alternative to training with annotated samples. GUI-Shift will be open-sourced at:
https://github.com/UbiquitousLearning/GUI-Shift.

## 1 INTRODUCTION

Mobile GUI agents (Gou et al., 2024; Hong et al., 2024; Qin et al., 2025; Wen et al., 2024; Yang et al., 2024) interpret natural language instructions and perform actions (e.g., click, scroll) directly on smartphone screens. They can control diverse apps as a human would, improving accessibility for users who are visually impaired, elderly, or have their hands occupied. Breakthroughs of vision language models (VLMs) (Bai et al., 2025; Chen et al., 2024; Xiaomi, 2025b) have reshaped the design paradigm of mobile GUI agents, transitioning from handcrafted heuristics to learned, vision-grounded policies. However, VLMs still struggle to deliver satisfactory accuracy (Dai et al., 2025; Qin et al., 2025; Rawles et al., 2024; Zhang et al., 2025a), especially when facing complex multi-step tasks. A common approach for enhancing VLMs is through supervised fine-tuning (SFT) on datasets containing GUI interaction trajectories paired with human-annotated task instructions (Li et al., 2024; Rawles et al., 2023). Yet effective, collecting GUI trajectories with task instructions remains labor-intensive and error-prone (Deka et al., 2017; Rawles et al., 2023). For example, the AndroidControl (Li et al., 2024) dataset takes one year of paid annotation effort to produce just 15,283 task demonstrations. Such high annotation cost limits the scalability of this paradigm.

In this study, we aim to address a fundamental challenge: *how to train capable mobile GUI agents using large-scale, unlabeled GUI trajectories, rather than relying on costly human-annotated instructions.* To tackle this, we propose a self-supervised training task, termed $K$-*step GUI Transition*. Inspired by inverse-dynamics modeling in robotics and biomechanics (Brandfonbrener et al., 2023; Tian et al., 2024; Zapolsky & Drumwright, 2017), where a model predicts control commands linking two consecutive physical states, our task treats screenshots as states and GUI actions as commands. Each training sample in $K$-step GUI Transition consists of two screenshots, $S_t$ and $S_{t+k}$, where

---

*Corresponding author.

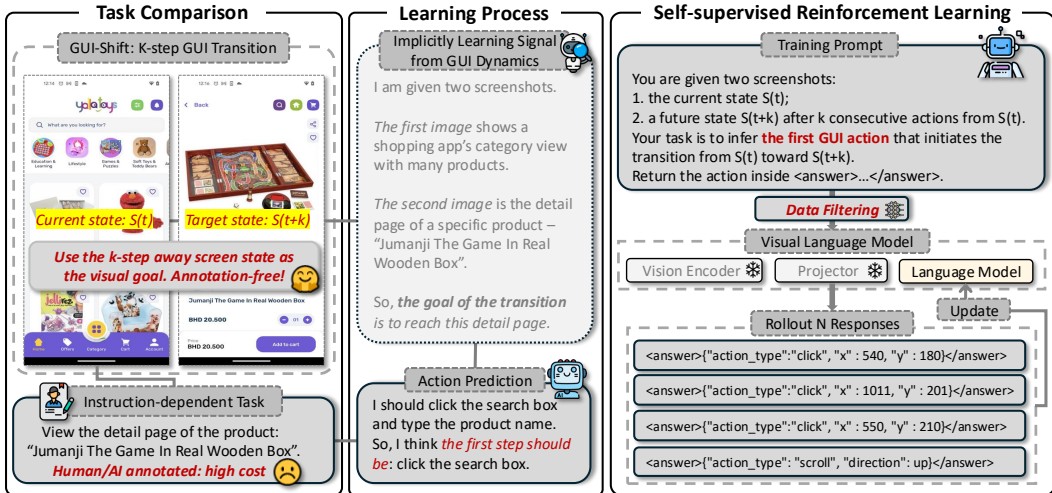

Figure 1: Overview of the GUI-Shift framework. **Left**: $K$-step GUI Transition replaces annotated instructions with the target state $S_{t+k}$, enabling scalable data construction through automated offline exploration. **Middle**: The model learns GUI dynamics by predicting the action that causes the transition. **Right**: GUI-Shift achieves self-supervised training by applying GRPO to GUI Transition.

$S_{t+k}$ results from executing $K$ actions starting from $S_t$. The VLM is trained to predict the first action that transforms $S_t$ into $S_{t+1}$. This design offers two key advantages: (1) *Explicit state-change signal.* Each sample contains a pair of GUI screenshots, enabling the model to utilize inter-screen visual differences and temporal cues, rather than learning from a single screen. (2) *Efficient data utilization at scale.* Since ground-truth actions are embedded in GUI trajectories (Rawles et al., 2023; Li et al., 2024), no predefined instructions or manual annotations are needed. Moreover, for any $k$, a GUI trajectory with $n$ screens can yield up to $n - k$ training samples, enabling scalable data construction. These benefits make $K$-step GUI Transition a strong candidate for self-supervised GUI agent training.

With the self-supervised training task, it is essential to determine how to effectively enhance VLMs. In GUI tasks, multiple action parameters can often be functionally equivalent and result in the same next state. For example, any coordinate within a button's bounding box is valid for a click, and textual inputs may be accepted in various formats or with different keywords. This multiplicity makes supervised fine-tuning (SFT) suboptimal, as it enforces a single reference action in a static dataset through the cross-entropy loss, penalizing all other valid alternatives and therefore providing misleading learning signals. To address this limitation, we adopt Group Relative Policy Optimization (GRPO) (Shao et al., 2024), which samples diverse plausible actions, evaluates them using a task-specific reward function, and ranks them based on group-normalized advantages. For example, in click actions, rewards are assigned if the sampled point lies within the target bounding box, offering a more tolerant and informative optimization signal. Overall, GRPO provides a more effective training paradigm for GUI agents by encouraging exploration and increasing robustness to action variability.

To this end, we present GUI-Shift, a self-supervised reinforcement learning (RL) framework that applies GRPO to $K$-step GUI Transition. Figure 1 illustrates the overview of the GUI-Shift framework. To select data matched to the model's learning ability, we adopt a unified action sampling and scoring mechanism during both data filtering and training stages. For each sample, the VLM generates $N$ action predictions, each scored based on format and action correctness. Only samples containing both correct and incorrect predictions among the $N$ predictions are selected for training. After training, the VLM acquires GUI-specific capabilities and serves as a more effective backbone for GUI agents. VLMs enhanced with GUI-Shift also have the ability to generalize well to GUI task automation and GUI grounding tasks without further alignment or fine-tuning.

We apply GUI-Shift to train four VLMs: Qwen2.5-VL-7B (Bai et al., 2025), InternVL3-8B (Chen et al., 2024), MimoVL-7B-SFT (Xiaomi, 2025b), and MimoVL-7B-RL, each using 2K samples for four $K$-step GUI Transition variants ($K \in \{1, 2, 3, 4\}$). We evaluate the VLMs on five benchmarks: AndroidControl (Li et al., 2024), GUI Odyssey (Lu et al., 2024), and AndroidWorld (Rawles et al.,

2024) for GUI task automation, and ScreenSpot-v2 (Wu et al., 2024b) and ScreenSpot-Pro (Li et al., 2025a) for GUI grounding. Overall, VLMs enhanced with GUI-Shift show notable improvements over their base versions. For example, GUI-Shift-Qwen achieves up to 11.2% higher accuracy on AndroidControl-High and 2.5% on ScreenSpot-v2, yielding 70.4% and 89.0% overall accuracy on the respective benchmarks. We also conduct comprehensive ablation studies to examine the effects of data filtering, task formulation, reasoning configurations, and training paradigms. Results show that using the target state $S_{t+k}$ as a visual instruction offers an effective alternative to human- or AI-annotated textual instructions for training GUI agents. Our contributions are summarized below.

(1) We introduce $K$-step GUI Transition, a training task that leverages abundant unlabeled GUI trajectories to enhance VLMs used in GUI agents.

(2) We propose GUI-Shift, a self-supervised RL framework that bridges the gap between GUI dynamics modeling and action-level GUI learning, mitigating the limitation of SFT in handling action multiplicity and poor generalization in GUI tasks.

(3) Experiments across four VLMs and five benchmarks show that VLMs enhanced with GUI-Shift exhibit generalization in both GUI automation and grounding tasks, with up to 11.2% accuracy gains.

## 2 RELATED WORK

### 2.1 MOBILE GUI AGENTS

Recent progress in mobile GUI agents has been driven by VLMs trained via SFT on large-scale datasets. These models learn to map instructions to GUI actions using instruction-following tasks, making high-quality annotations essential. Despite the availability of diverse GUI datasets (Deka et al., 2017; Gao et al., 2024; Li et al., 2024; Lu et al., 2024; Rawles et al., 2023), the quantity of high-quality annotations remains insufficient for robust training and usually requires significant human effort to scale. To reduce annotation costs, prior pipelines often incorporate out-of-domain image-caption pairs (Hong et al., 2024; Wang et al., 2024) and supplement training with web and desktop data to improve cross-platform generalization (Cheng et al., 2024). As a result, the overall scale of training data tends to be large: Uground (Gou et al., 2024) uses 1.3M screenshots to train a visual grounding model; OS-Atlas (Wu et al., 2024b) leverages 13M GUI elements for grounding pretraining. Some recent approaches have explored GUI state modeling. UI-TARS (Qin et al., 2025) incorporates a state transition task, which focuses on describing visual changes between screenshots rather than predicting the underlying actions, resulting in a gap with GUI task automation. MobileVLM (Wu et al., 2024a) introduces an action prediction task between screenshots, but is restricted to one-step transitions and SFT. They still rely on annotation-heavy fine-tuning for downstream alignment and generalization. In this work, we propose $K$-step GUI Transition, formulating a $k$-step inverse dynamics objective that enables scalable training on large, unlabeled, and underutilized GUI datasets.

### 2.2 RULE-BASED REINFORCEMENT LEARNING

Rule-based RL has proven to be a promising alternative to SFT. GRPO (Shao et al., 2024) uses a reward model to score each response and computes group relative advantages instead of training a critic model, whose size is comparable to the policy model, thereby significantly reducing computational cost. Reinforcement Learning with Verifiable Rewards (Lambert et al., 2024) further emphasize the use of verifiable answers to design reliable reward signals. DeepSeek-R1 (Guo et al., 2025) shows that simple format and accuracy rewards are sufficient to surpass the performance of instruction-tuned models. Several recent works have applied GRPO on GUI tasks: UI-R1 (Lu et al., 2025) employs one-stage RL on 136 samples with step-level instructions. GUI-R1 (Xia & Luo, 2025) expands this to 3K task-level instructions from five platforms. InfiGUI-R1 (Liu et al., 2025) adopts a two stage SFT+RL pipeline and scales to 32K samples from both GUI and non-GUI domains. UI-Venus (Gu et al., 2025) employs GRPO to two variants, using 107K samples for grounding and 350K for navigation. While these works demonstrate the effectiveness of rule-based RL for GUI agents, they still rely on annotated instructions and require reasoning during training and inference. Different from these annotation-dependent training paradigms, GUI-Shift fine-tunes VLMs via one-stage RL on $K$-step GUI Transition in a self-supervised manner, achieving competitive performance and demonstrating strong generalization across GUI task automation and GUI grounding benchmarks.

## 3 METHODOLOGY

GUI-Shift is a self-supervised RL framework designed to enhance VLM-based GUI agents through the $K$-step GUI Transition task. In this section, we first describe GRPO, the underlying training algorithm in GUI-Shift. We then detail the reward design tailored to GUI action modeling, and present the complete GUI-Shift framework along with its rationale and advantages.

### 3.1 PRELIMINARIES

GRPO (Shao et al., 2024) offers a computationally efficient alternative to Proximal Policy Optimization (PPO) (Schulman et al., 2017), a widely used actor-critic method. Instead of maintaining a separate critic network for value estimation, GRPO computes normalized, group-wise advantages $A_i$ directly from reward scores, thereby removing the value function update and lowering computational cost. The GRPO objective in our framework is defined as follows:

$$\mathcal{J}_{\text{GRPO}}(\theta) = \mathbb{E}\left[q \sim P(Q), \{o_i\}_{i=1}^{G} \sim \pi_{\theta_{old}}(O \mid q)\right]$$

$$\frac{1}{G}\sum_{i=1}^{G}\left(\min\left(\rho_i A_i, \, \text{clip}\left(\rho_i, \, 1-\epsilon, \, 1+\epsilon\right) A_i\right) - \beta \, \mathbb{D}_{\text{KL}}(\pi_\theta \| \pi_{\text{ref}})\right), \tag{1}$$

$$\text{where} \quad \rho_i = \frac{\pi_\theta(o_i \mid q)}{\pi_{\theta_{old}}(o_i \mid q)}, \quad A_i = \frac{r_i - \text{mean}(\{r_1, r_2, \cdots, r_G\})}{\text{std}(\{r_1, r_2, \cdots, r_G\})}.$$

Specifically, for each question $q$ in the training set, we sample a group of outputs $\{o_1, o_2, \ldots, o_G\}$ from the old policy $\pi_{\theta_{old}}$ using high temperature decoding, and compute the group-wise relative advantage $A_i$ for each output. A clipped surrogate objective, along with a KL divergence regularizer toward the reference policy $\pi_{\text{ref}}$, is then used to update model parameters and ensure training stability.

### 3.2 REWARD DESIGN

The reward function plays a central role in guiding and stabilizing model optimization. In GUI action prediction, each answer is a structured action comprising a verifiable action type and associated parameters, making the task well-suited to a rule-based reward formulation. Following DeepSeek-R1 (Guo et al., 2025), we adopt a rule-based reward $R$ tailored for GUI tasks, which combines a format reward $R_f$ to enforce output consistency and an action reward $R_a$ to evaluate action correctness:

$$R = R_f + R_a \tag{2}$$

**Format reward.** To ensure that model outputs are well-structured and easy to parse, GUI-Shift requires the final answer to be enclosed in `<answer>...</answer>` tags during training. Predictions conforming to the expected format receive $R_f = 1$; otherwise, $R_f = 0$. Unlike prior methods (Lu et al., 2025; Liu et al., 2025; Xia & Luo, 2025), GUI-Shift omits explicit reasoning traces in outputs, eliminating reasoning token generation and substantially reduces training time. For example, training Qwen2.5-VL-7B on 2K $K$-step GUI Transition samples requires only 9 hours, compared to 17 hours with reasoning traces, without compromising downstream performance, as shown in Table 4.

**Action reward.** We adopt a unified action space of eight types for both training and inference. The action space comprises eight types, each as a JSON object with *action_type* and type-specific parameters: *click* and *long_press* require a target point; *scroll* requires a direction; *open_app* requires an app name; *input_text* requires the input content; and *navigate_back*, *navigate_home* and *wait* require no parameters. The action reward $R_a$ is defined accordingly:

$$R_a = \begin{cases} 1, & \text{if } x_1 \leq \hat{x} \leq x_2 \text{ and } y_1 \leq \hat{y} \leq y_2, \quad t \in \{click, long\_press\}; \\ 1, & \text{if } \hat{t} = t \text{ and } \hat{p} = p, \quad t \in \{open\_app, input\_text, scroll\}; \\ 1, & \text{if } \hat{t} = t, \quad t \in \{navigate\_back, navigate\_home, wait\}; \\ 0, & \text{otherwise.} \end{cases} \tag{3}$$

Here, $\hat{t}$ and $\hat{p}$ denote the predicted action type and parameter, $t$ and $p$ denote their ground-truth counterparts; $\hat{x}, \hat{y}$ are the predicted coordinates, and $[x_1, y_1, x_2, y_2]$ is the ground-truth bounding box.

### 3.3 GUI-SHIFT FRAMEWORK

$K$**-step GUI Transition.** While existing VLMs can parse individual GUI screens due to exposure to GUI data during pretraining, they still lack the temporal reasoning capabilities required for complex multi-step GUI tasks. To bridge this gap, we propose the $K$-step GUI Transition task, which asks the model to predict the first action that transitions a given state $S_t$ to a future state $S_{t+k}$, as shown in Figure 1. Compared to annotated approaches, our task offers two key advantages. First, while annotated tasks require costly and error-prone textual annotations for each step, $K$-step GUI Transition leverages state pairs directly extracted from GUI trajectories. The future state $S_{t+k}$, obtained after executing $K$ actions from $S_t$, serves as an explicit visual goal, providing a supervision signal that is not only annotation-free but also more concrete and informative than textual instructions. Second, rather than mapping textual instructions to actions, our task compels the model to interpret and compare both the current and target GUI states, infer the transition goal, and identify the action that initiates the state change. Overall, by leveraging visual goals and requiring temporal reasoning across state pairs, this more challenging formulation fosters a deeper understanding of GUI dynamics and provides a scalable, practical solution for robust GUI agent training.

**Self-supervised RL.** During training, for each sample, the model generates $N$ candidate actions ($N = 8$ in our experiments), each evaluated by a rule-based reward that integrates format and action correctness, as detailed in Section 3.2. Group-wise normalized advantages are then computed, and optimization proceeds as outlined in Section 3.1. GRPO is particularly well-suited to GUI-Shift for three reasons: (1) Compared to PPO, it eliminates the need for a separate value function, typically in the same size as the policy model. This substantially reduces computational overhead and better supports our efficiency objectives; (2) Compared to SFT, it enables flexible reward assignment tailored to each action type. For instance, click actions are considered correct if the predicted point falls within the ground-truth bounding box rather than requiring an exact match in SFT, which better reflects practical GUI grounding requirements; (3) The $N$-candidate sampling mechanism encourages exploration and model can learn from optimal candidates while avoiding suboptimal ones.

**Data filtering pipeline.** To prepare high-quality $K$-step GUI Transition data, we perform data filtering using the same action sampling and scoring mechanism as in training. First, for each $K \in \{1, 2, 3, 4\}$, we construct a pool of candidate state pairs $(S_t, S_{t+k})$ from the original dataset. Next, for each pair, the model generates 8 responses using the same sampling temperature as in GRPO training. Each response is then evaluated with the reward function described in Section 3.2. Finally, we retain only those pairs with both correct and incorrect responses. By applying this filtering process to each model independently, the final training set is both challenging and informative, and well aligned with the model's learning capacity.

Taken together, these design choices enable GUI-Shift to provide higher efficiency from three aspects: (1) *Scalable data construction.* Without relying on annotated instructions, GUI-Shift enables large-scale filtering of training data at minimal cost. For example, for Mimo-VL-7B-RL, we filtered 2,920 high-quality samples out of 8K original 1-step GUI Transition pairs, without any annotation waste. (2) *Maximized data utilization.* For each $K$, an $N$-image trajectory can yield up to $N - K$ training pairs, maximizing data utilization for fixed-length GUI trajectories. (3) *Reduced training cost.* Without explicit reasoning traces during training, GUI-Shift avoids extra token decoding and reduces training time by nearly 50%, from 17 to 9 hours on 2K samples under our experimental setup.

## 4 EXPERIMENTS

In this section, we first detail the experimental setup, including data construction and training configurations (Section 4.1). We then evaluate models trained with $K$-step GUI Transition ($k \in \{1, 2, 3, 4\}$) on static GUI task automation and grounding benchmarks (Section 4.2) as well as end-to-end GUI task automation benchmarks (Section 4.3). Finally, we conduct comprehensive ablation studies to analyze key design choices from four perspectives: data filtering, task formulation, reasoning configurations during RL, and training algorithms (Section 4.4).

### 4.1 EXPERIMENTAL SETUP

**Training configurations.** Using the open-source VLM-R1 (Shen et al., 2025) framework, we fine-tune Qwen2.5-VL-7B (Bai et al., 2025), InternVL3-8B (Chen et al., 2024), MimoVL-7B-

SFT (Xiaomi, 2025b), and MimoVL-7B-RL with the pipeline described in Section 3. During training, only the language model is optimized while the vision encoder and projector are frozen. All experiments are conducted on 8×NVIDIA H100 GPUs. Hyper-parameters are listed in Appendix B.

**Data construction.** All training data are sourced from the training set of AndroidControl (Li et al., 2024), which provides GUI trajectories paired with human-labeled instructions. These instructions enable both self-supervised GUI-Shift training and comparisons with VLMs trained using SFT (see Section 4.4 for a comparison of the two training approaches). Following the data filtering pipeline discussed in Section 3, we select 2K samples per $K$ for each model. For Qwen2.5-VL-7B, the proportion of samples with either entirely correct or entirely incorrect actions was exceptionally high. As a result, we use unfiltered data for its training.

## 4.2 STATIC BENCHMARKS AND RESULTS

**GUI task automation.** We evaluate GUI-Shift on two task automation benchmarks: *AndroidControl* (Li et al., 2024) and *GUI Odyssey* (Lu et al., 2024). AndroidControl provides two test settings: *AndroidControl-Low*, which assesses step-level instruction following ability (e.g., "Type the product name in the search box"), and *AndroidControl-High*, which evaluates long-horizon task planning (e.g., "View the detail page of the product"). GUI Odyssey offers a more challenging evaluation, encompassing both phone and tablet applications as well as cross-app scenarios. The test set includes 9,134 samples in AndroidControl and 27,493 in GUI Odyssey, covering six action types: *click*, *long_press*, *scroll*, *navigate_back*, *navigate_home*, and *input_text*. We report *type match* (TM) that represents the proportion of samples with the correct action type, and *exact match* (EM), which requires both the action type and all parameters to be correct. Metrics are computed using AgentCPM-GUI (Zhang et al., 2025b) and GUIEvalKit (Xiaomi, 2025a).

Table 1: Performance comparison on GUI task automation benchmarks: AndroidControl (AC-Low, AC-High) and GUI Odyssey. GUI-Shift achieves substantial improvements over base models. **Bold**: the best result; underlined: the second best result. TM: type match; EM: exact match.

| Model | # Training Samples | AC-Low TM | AC-Low EM | AC-High TM | AC-High EM | GUI Odyssey TM | GUI Odyssey EM |
|---|---|---|---|---|---|---|---|
| *Proprietary models* | | | | | | | |
| GPT-4o (OpenAI, 2024) | - | 74.3 | 19.4 | 66.3 | 20.8 | 34.3 | 3.3 |
| *Models trained with annotations* | | | | | | | |
| SeeClick (Cheng et al., 2024) | 1M | 93.0 | 75.0 | 82.9 | 59.1 | 71.0 | 53.9 |
| OS-Atlas-7B (Wu et al., 2024b) | 2.3M | 93.6 | 85.2 | 85.2 | 71.2 | - | 62.0 |
| Aguvis-7B (Xu et al., 2024) | 1M | - | 80.5 | - | 61.5 | - | - |
| UI-TARS-7B (Qin et al., 2025) | - | 98.0 | 90.8 | 83.7 | 72.5 | **94.6** | **87.0** |
| UI-R1-3B (Lu et al., 2025) | 136 | 94.3 | 88.5 | 57.9 | 45.4 | 52.2 | 32.5 |
| GUI-R1-7B (Xia & Luo, 2025) | 3K | 85.2 | 66.5 | 71.6 | 51.7 | 65.5 | 38.8 |
| InfiGUI-R1-3B (Liu et al., 2025) | 32K | 96.0 | 92.1 | 82.7 | 71.1 | - | - |
| AgentCPM-GUI (Liu et al., 2025) | 470K | 94.4 | 90.2 | 77.7 | 69.2 | 90.9 | 75.0 |
| UI-Venus-Navi-7B (Gu et al., 2025) | 350K | 97.1 | 92.4 | 86.5 | **76.1** | 87.3 | 71.5 |
| *Ours: Qwen2.5-VL-7B as the base model* | | | | | | | |
| Qwen2.5-VL-7B (Bai et al., 2025) | - | 94.9 | 83.8 | 72.9 | 59.2 | 59.8 | 44.9 |
| GUI-Shift-Qwen ($k=1$) | 2K | 98.0↑3.1 | 90.6↑6.8 | 85.9↑13.0 | 70.4↑11.2 | 78.5↑18.7 | 54.8↑9.9 |
| *Ours: InternVL3-8B as the base model* | | | | | | | |
| InternVL3-8B (Chen et al., 2024) | - | 97.8 | 90.0 | 71.5 | 49.8 | 48.8 | 20.3 |
| GUI-Shift-Intern ($k=4$) | 2K | 97.3↓0.5 | 88.0↓2.0 | 78.5↑7.0 | 56.6↑6.8 | 59.6↑10.8 | 23.3↑3.0 |
| *Ours: Mimo-VL-7B-SFT as the base model* | | | | | | | |
| Mimo-VL-7B-SFT (Xiaomi, 2025b) | - | 90.8 | 85.7 | 75.2 | 63.1 | 86.9 | 62.0 |
| GUI-Shift-Mimo-SFT ($k=3$) | 2K | 98.6↑7.8 | **93.2**↑7.5 | **87.2**↑12.0 | 73.4↑10.3 | 86.1↓0.8 | 60.7↓1.3 |
| *Ours: Mimo-VL-7B-RL as the base model* | | | | | | | |
| Mimo-VL-7B-RL (Xiaomi, 2025b) | - | 91.8 | 87.2 | 76.5 | 64.6 | 87.2 | 63.1 |
| GUI-Shift-Mimo-RL ($k=1$) | 2K | **98.9**↑7.1 | **93.2**↑6.0 | 86.9↑10.4 | 71.7↑7.1 | 84.8↓2.4 | 59.5↓3.6 |

● *GUI-Shift generally improves performance over base models on GUI task automation benchmarks.*
Table 1 presents the results of GUI-Shift, together with comparisons against both base models and
models trained with annotations. GUI-Shift achieves notable gains across all four models, especially
on AndroidControl-High. GUI-Shift-Qwen raises EM by 11.2% over Qwen2.5-VL-7B, while GUI-
Shift-Mimo-SFT and GUI-Shift-Mimo-RL reach gains of 10.3% and 7.1%, respectively. On GUI
Odyssey, minor declines for GUI-Shift-Mimo-SFT and GUI-Shift-Mimo-RL likely result from
the 1,381 tablet episodes in the test set, whose GUI layouts differ substantially from smartphones.
Compared to models trained with annotations, GUI-Shift achieves comparable or even superior results
on all benchmarks using only 2K $K$-step GUI Transition samples. Overall, these results underscore
the robustness and effectiveness of our approach in GUI task automation.

**GUI grounding.** We evaluate GUI-Shift on two GUI grounding benchmarks: *ScreenSpot-v2* (Wu
et al., 2024b) with 1,272 samples from mobile, desktop, and web platforms, and *ScreenSpot-Pro* (Li
et al., 2025a) with 1,581 high-resolution screenshots for fine-grained evaluation. Evaluations for the
base models and GUI-Shift are adapted from ScreenSpot-Pro-GUI-Grounding (Li et al., 2025b).

● *GUI-Shift consistently outperforms base models and surpasses most existing baselines on GUI
grounding benchmarks.* Table 2 summarizes the overall and baseline results. Across all models,
GUI-Shift delivers improved accuracy over base models, with the best variants reaching 2.5% and
1.5% gains on ScreenSpot-v2 and ScreenSpot-Pro, respectively. Moreover, GUI-Shift surpasses all
annotation-trained models except UI-Venus-Ground-7B, which is trained specifically for the GUI
grounding using 107K annotated samples. These results demonstrate that models trained solely on
unlabeled GUI Transition data can effectively transfer to challenging GUI grounding tasks. [1]

Table 2: Performance comparison on GUI grounding benchmarks: ScreenSpot-v2 and ScreenSpot-
Pro. GUI-Shift exhibits strong generalization and achieves the second best result on ScreenSpot-Pro.
**Bold**: the best result; underlined: the second best result.

| Model | # Training Samples | ScreenSpot-v2 | | | | | | | -Pro |
| | | Mobile | | Desktop | | Web | | Avg. | Avg. |
| | | Text | Icon | Text | Icon | Text | Icon | | |
| --- | --- | --- | --- | --- | --- | --- | --- | --- | --- |
| *Models trained with annotations* | | | | | | | | | |
| CogAgent-18B (Wang et al., 2024) | - | - | - | - | - | - | - | - | 7.7 |
| SeeClick-9.6B (Cheng et al., 2024) | 1M | 78.4 | 50.7 | 70.1 | 29.3 | 55.2 | 32.5 | 55.1 | 1.1 |
| UGround-7B (Gou et al., 2024) | 1.3M | 75.1 | 84.5 | 85.1 | 61.4 | 84.6 | 71.9 | 76.3 | 16.5 |
| OS-Atlas-7B (Wu et al., 2024b) | 2.3M | 95.2 | 75.8 | 90.7 | 63.6 | 90.6 | 77.3 | 84.1 | 18.9 |
| ShowUI-2B (Lin et al., 2024) | 256K | - | - | - | - | - | - | - | 7.7 |
| UI-TARS-7B (Qin et al., 2025) | - | 96.9 | 89.1 | 95.4 | 85.0 | 93.6 | 85.2 | 91.6 | 35.7 |
| UI-R1-E-3B (Lu et al., 2025) | 2K | 83.0 | **97.1** | 85.0 | 91.7 | 77.9 | **95.4** | 89.5 | 33.5 |
| InfiGUI-R1-3B (Liu et al., 2025) | 32K | - | - | - | - | - | - | - | 35.7 |
| LPO (Tang et al., 2025) | - | 97.9 | 82.9 | 95.9 | 86.4 | 95.6 | 84.2 | 90.5 | - |
| UI-Venus-Ground-7B (Gu et al., 2025) | 107K | **99.0** | 90.0 | 97.0 | 90.7 | **96.2** | 88.7 | **94.1** | **50.8** |
| *Ours: Qwen2.5-VL-7B as the base model* | | | | | | | | | |
| Qwen2.5-VL-7B (Bai et al., 2025) | - | 98.3 | 86.3 | 88.7 | 67.1 | 92.7 | 81.8 | 87.7 | 26.4 |
| GUI-Shift-Qwen ($k = 4$) | 2K | 98.6 | 89.6 | 86.1 | 75.0 | 92.7 | 82.8 | 89.0↑1.3 | 27.1↑0.7 |
| *Ours: InternVL3-8B as the base model* | | | | | | | | | |
| InternVL3-8B (Chen et al., 2024) | - | 93.4 | 81.5 | 80.4 | 52.1 | 91.0 | 73.4 | 81.3 | 15.0 |
| GUI-Shift-Intern ($k = 1$) | 2K | 93.8 | 83.4 | 80.4 | 51.4 | 91.0 | 73.4 | 81.6↑0.3 | 15.4↑0.4 |
| *Ours: Mimo-VL-7B-SFT as the base model* | | | | | | | | | |
| Mimo-VL-7B-SFT (Xiaomi, 2025b) | - | 96.6 | 84.4 | 92.8 | 80.0 | 88.9 | 76.8 | 87.6 | 39.8 |
| GUI-Shift-Mimo-SFT ($k = 1$) | 2K | 98.3 | 87.7 | 92.3 | 82.1 | 94.0 | 79.8 | 90.1↑2.5 | 40.7↑0.9 |
| *Ours: Mimo-VL-7B-RL as the base model* | | | | | | | | | |
| Mimo-VL-7B-RL (Xiaomi, 2025b) | - | 98.3 | 86.3 | 90.2 | 80.7 | 92.7 | 75.4 | 88.4 | 40.2 |
| GUI-Shift-Mimo-RL ($k = 1$) | 2K | **99.0** | 87.7 | 91.2 | 83.6 | 89.7 | 72.9 | 88.4↑0.0 | 41.7↑1.5 |

---

[1]Detailed results for different $K$ values are provided for GUI task automation and GUI grounding benchmarks
in Appendix C and Appendix D, respectively.

### 4.3 END-TO-END BENCHMARKS AND RESULTS

**Evaluation setup.** Beyond the single-step evaluation on AndroidControl described in Section 4.2, we additionally evaluate end-to-end performance. An episode is considered successful only if all steps are correct in both action type and parameters; otherwise, the entire trajectory is counted as a failure. This stricter metric better reflects realistic GUI task automation. We also evaluate GUI-Shift on AndroidWorld (Rawles et al., 2024), a more challenging interactive benchmark consisting of 116 tasks. We follow the official M3A agent protocol, where the model receives screenshots annotated with set-of-mark (Yang et al., 2023) and generates one action and a summary at each step. We evaluate Mimo-VL-7B-SFT and GUI-Shift-Mimo-SFT ($K \in \{1, 2, 3, 4\}$) under this configuration.

Table 3: End-to-end performance on AndroidControl (AC-Low, AC-High) across base models. GUI-Shift improves over the base models in most cases. **Bold**: the best result.

| Model | Qwen2.5-VL-7B | | InternVL3-8B | | Mimo-VL-7B-SFT | | Mimo-VL-7B-RL | |
|---|---|---|---|---|---|---|---|---|
| | AC-Low | AC-High | AC-Low | AC-High | AC-Low | AC-High | AC-Low | AC-High |
| Base | 50.2 | 22.4 | 68.3 | 15.4 | 48.4 | 16.4 | 53.3 | 18.0 |
| K=1 | 67.5↑17.3 | 29.9↑7.5 | 60.7↓7.6 | 18.9↑3.5 | 72.4↑24.0 | 32.1↑15.7 | **76.3**↑23.0 | 32.2↑14.2 |
| K=2 | 65.1↑14.9 | 29.9↑7.5 | 60.3↓8.0 | 19.2↑3.8 | 73.9↑25.5 | 33.1↑16.7 | 71.8↑18.5 | **34.6**↑16.6 |
| K=3 | **69.3**↑19.1 | 29.4↑7.0 | 61.0↓7.3 | 19.2↑3.8 | **75.7**↑27.3 | **34.1**↑17.7 | 71.0↑17.7 | 33.6↑15.6 |
| K=4 | 67.6↑17.4 | **28.7**↑6.3 | **61.1**↓7.2 | **21.9**↑6.5 | 74.2↑25.8 | 31.1↑14.7 | 69.5↑16.2 | 33.7↑15.7 |

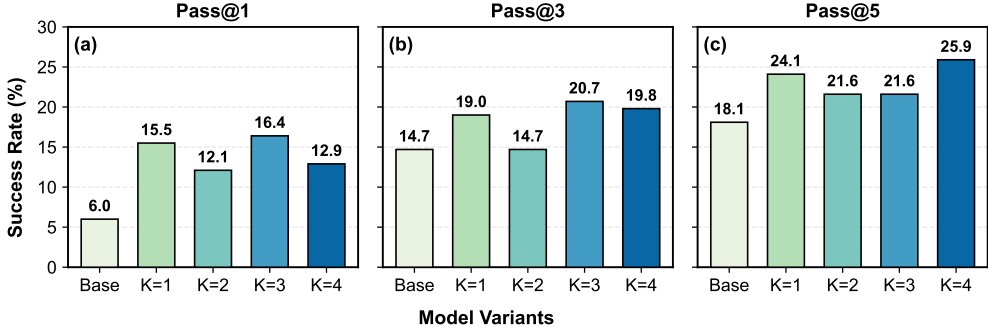

Figure 2: Performance on AndroidWorld of the base model Mimo-VL-7B-SFT and GUI-Shift-Mimo-SFT with different $K$. Evaluation follows the original M3A agent protocol. GUI-Shift consistently improves success rates across Pass@1, Pass@3, and Pass@5.

● *GUI-Shift improves end-to-end performance across benchmarks.* Table 3 presents end-to-end performance on AndroidControl, where GUI-Shift improves over the base models in most cases. For example, Mimo-VL-7B-SFT improves from 48.4% to 75.7% on AndroidControl-Low and from 16.4% to 34.1% on AndroidControl-High. Mimo-VL-7B-RL and Qwen2.5-VL-7B also show gains across both settings. For InternVL3-8B, performance decreases on AndroidControl-Low compared to the base model, while increasing on the more challenging AndroidControl-High from 15.4% to 21.9%. Figure 2 illustrates consistent improvements on AndroidWorld under Pass@1, Pass@3, and Pass@5. GUI-Shift-Mimo-SFT increases the success rate from 6.0% to 16.4% for $K = 3$ at Pass@1, and from 18.1% to 25.9% for $K = 4$ at Pass@5. These results indicate that GUI-Shift improves performance not only in single-step accuracy but also in end-to-end performance, demonstrating its potential for real-world GUI task automation.

### 4.4 ABLATION STUDY

**Data filtering.** We evaluate InternVL3-8B, MimoVL-7B-SFT, and MimoVL-7B-RL trained with and without data filtering on both GUI task automation and GUI grounding benchmarks. For each $K$-step GUI Transition sample, each model generates 8 responses to build its candidate pool, retaining only those samples where predictions are both correct and incorrect. We select 2K training samples per model and $k$ from each filtered pool and the original training set, respectively.

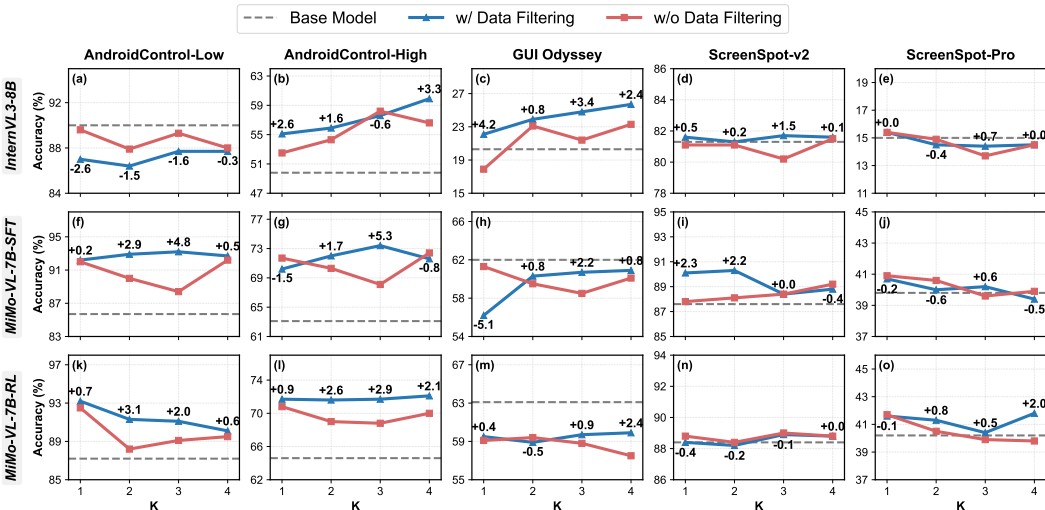

Figure 3: Impact of Data filtering. Each model is fine-tuned on 2K $K$-step GUI Transition samples. Filtered data are more informative and challenging, and outperform unfiltered ones.

• *Data filtering improves accuracy on both GUI task automation and GUI grounding benchmarks.* Figure 3 shows that models trained with filtered data achieve higher accuracy than those trained on unfiltered data in most cases. For example, on AndroidControl-Low, Mimo-VL-7B-SFT achieves up to 4.8% higher accuracy (Figure 3(f), $K = 3$), and on ScreenSpot-v2, up to 2.3% (Figure 3(i), $K = 1$). These results suggest that our filtering mechanism effectively selects more informative and challenging samples for GUI agent training. Moreover, since $K$-step GUI Transition does not require human-annotated instructions, this filtering process scales easily and incurs minimal cost.

**Task formulation.** We compare $K$-step GUI Transition with two annotated baselines. To ensure fairness, we do not apply data filtering, and all models in each comparison are trained on the same set of 2K samples, with identical current states $S_t$ and ground-truth actions. The only difference is the instruction type: baselines pair $S_t$ with a human-annotated *task instruction* or *step instruction*, while $K$-step GUI Transition uses the target state $S_{t+k}$ as the visual instruction.

• *Using $S_{t+k}$ as the visual target outperforms using textual instructions as input.* Table 4 shows that VLMs trained with $K$-step GUI Transition achieves better performance than those with annotated tasks in most cases. For example, on AC-Low and AC-High, InternVL3-8B trained with GUI Transition achieves 4.0% and 3.6% higher EM accuracy, respectively, than when trained with task instructions. Qwen2.5-VL-7B also achieves the highest EM accuracy with GUI Transition across all benchmarks. These results indicate that $S_{t+k}$ provides a more informative signal than human-annotated instructions.

**Reasoning configurations.** To verify the effect of reasoning during training, we compare models fine-tuned on 2K $K$-step GUI Transition data with and without `<think>...</think>`.

• *Excluding reasoning boosts performance and efficiency for GUI-Shift.* Table 4 shows that omitting explicit reasoning requirements during training not only maintains but often improves performance. For InternVL3-8B, training without reasoning achieves up to 7.9% higher EM on AndroidControl-High; Qwen2.5-VL-7B shows consistent gains of about 2% across benchmarks. Moreover, removing reasoning nearly halves training time cost, reducing it from 17 to 9 hours for Qwen2.5-VL-7B and from 15 to 7 hours for InternVL3-8B. These results confirm that excluding reasoning both improves performance and significantly enhances training efficiency.

**Training algorithms.** For each $K \in \{1, 2, 3, 4\}$, we fine-tune Qwen2.5-VL-7B and InternVL3-8B on 2K identical $K$-step GUI Transition data using SFT or GRPO, and evaluate them on AndroidControl.

• *GRPO is more suitable than SFT for $K$-step GUI Transition.* Figure 4 shows that GRPO improves accuracy in most cases, whereas SFT consistently reduces accuracy compared to both the base models and GRPO, with drops up to 65.1% relative to GRPO (Figure 4(c), $K = 3$). We attribute this to its

Table 4: Performance on GUI task automation under different training settings. GUI-Shift outperforms models trained with textual instructions or with explicit reasoning requirements. **Bold**: the best result. TM: type match; GR: grounding accuracy for clicks; EM: exact match.

| Model | AndroidControl-Low | | | AndroidControl-High | | | GUI Odyssey | | |
|---|---|---|---|---|---|---|---|---|---|
| | TM | GR | EM | TM | GR | EM | TM | GR | EM |
| *Base model: Qwen2.5-VL-7B* | | | | | | | | | |
| Qwen2.5-VL-7B | 94.9 | 90.9 | 83.8 | 72.9 | 66.6 | 59.2 | 59.8 | 47.5 | 44.9 |
| + *Task Instruction* | 97.9↑3.0 | 93.5↑2.6 | 90.5↑6.7 | 85.3↑12.4 | 76.2↑9.6 | 69.9↑10.7 | 74.1↑14.3 | 62.0↑14.5 | 51.8↑6.9 |
| + *Step Instruction* | 97.7↑2.8 | 93.7↑2.8 | 86.4↑2.6 | 82.4↑9.5 | 73.1↑6.5 | 67.2↑8.0 | 74.5↑14.7 | 62.7↑15.2 | 51.5↑6.6 |
| *Ours (w/ reasoning)* | 95.5↑0.6 | 91.1↑0.2 | 88.2↑4.4 | 83.8↑10.9 | 75.6↑9.0 | 69.0↑9.8 | 74.0↑14.2 | 63.5↑16.0 | 51.6↑6.7 |
| ***Ours*** | **98.0**↑3.1 | **94.0**↑3.1 | **90.6**↑6.8 | **85.9**↑13.0 | **77.5**↑10.9 | **70.4**↑11.2 | **78.5**↑18.7 | **67.2**↑19.7 | **54.8**↑9.9 |
| *Base model: InternVL3-8B* | | | | | | | | | |
| InternVL3-8B | 97.8 | 92.4 | 90.0 | 71.5 | 54.6 | 49.8 | 48.8 | 20.2 | 20.3 |
| + *Task Instruction* | 95.9↓1.9 | **92.8**↑0.4 | 85.3↓4.7 | 79.7↑8.2 | 65.8↑11.2 | 54.6↑4.8 | 57.3↑8.5 | 31.5↑11.3 | 26.5↑6.2 |
| + *Step Instruction* | 96.8↓1.0 | 92.8↑0.4 | 86.0↓4.0 | **80.7**↑9.2 | **66.0**↑11.4 | 54.5↑4.7 | **62.6**↑13.8 | **34.4**↑14.2 | **28.8**↑8.5 |
| *Ours (w/ reasoning)* | 97.3↓0.5 | 92.3↑0.1 | 87.8↓2.2 | 72.4↑0.9 | 55.8↑1.2 | 50.3↑0.5 | 38.9↓9.9 | 18.8↓1.4 | 16.2↓4.1 |
| ***Ours*** | 97.5↓0.3 | 92.6↑0.2 | **89.3**↓0.7 | 78.6↑7.1 | 61.7↑7.1 | **58.2**↑8.4 | 51.4↑2.6 | 21.4↑1.2 | 21.4↑1.1 |

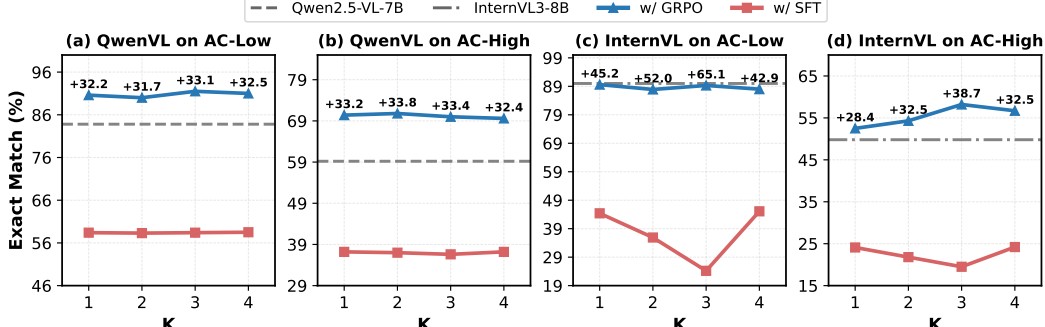

Figure 4: Comparison of training algorithms for the $K$-step GUI Transition task ($k \in \{1, 2, 3, 4\}$). Qwen2.5-VL-7B and InternVL3-8B are fine-tuned with 2K samples for each $k$ and evaluated on AndroidControl. GRPO provides notable performance gains over SFT for all models and settings.

sensitivity to format mismatch between training and evaluation. These results confirm that GRPO is a more effective choice for transferring $K$-step GUI Transition knowledge to GUI task automation.

## 5 CONCLUSION

This study introduces GUI-Shift, a self-supervised reinforcement learning framework for training VLM-based GUI agents without relying on costly annotations. Based on the $K$-step GUI Transition training task, enhancing VLMs with GUI-Shift captures temporal dynamics between GUI states and provides a scalable, annotation-free training signal. Experiments across four VLMs and five benchmarks show consistent improvements, including up to 11.2% gains in GUI task automation accuracy and strong generalization to GUI grounding tasks. These results demonstrate that self-supervised RL can effectively exploit unlabeled GUI trajectories, offering a practical and efficient alternative to training tasks with human- or AI-annotated textual instructions.

## ACKNOWLEDGMENTS

This work was supported by the National Natural Science Foundation of China (No. 62522202) and the Beijing Natural Science Foundation (No. L253005).

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

## A    USE OF LARGE LANGUAGE MODELS (LLMS)

We use LLMs for writing polishing and language refinement.

## B    TRAINING HYPER-PARAMETERS

The hyper-parameter details for GUI-Shift are provided in Table 5.

Table 5: Hyper-parameter settings used for all GRPO training.

| Hyper-parameter | Value |
| --- | --- |
| learning_rate | from 1e-6 to 0 |
| temperature | 0.9 |
| num_generations | 8 |
| num_train_epochs | 4 |
| max_prompt_length | 1024 |
| max_completion_length | 256 |
| per_device_train_batch_size | 2 |
| gradient_accumulation_steps | 8 |
| $\epsilon$ (clipping parameter) | 0.2 |
| $\beta$ (KL coefficient) | 0.04 |

## C    Task Automation Results with and without Data Filtering

Table 6 and Table 7 report the performance of GUI-Shift on AndroidControl and GUI Odyssey benchmarks using 2K $K$-step GUI Transition samples, with and without data filtering. Without data filtering, as shown in Table 6, GUI-Shift trained with each $k$ value outperforms the base models on most benchmarks. Specifically, for Qwen2.5-VL-7B, GUI-Shift delivers consistent improvements across all benchmarks; for the other three models, GUI-Shift also improves task automation in most cases, especially on AndroidControl. These results demonstrate that conditioning on the future state $S_{t+k}$ consistently provides an effective supervision signal across different transition step sizes. We apply data filtering is to InternVL3-8B, MimoVL-7B-SFT, and MimoVL-7B-RL. With data filtering, as shown in Table 7, model performance further improves in most settings, confirming that filtering enhances data quality and strengthens VLM optimization.

Table 6: Performance on task automation benchmarks: AndroidControl and GUI Odyssey. Models are fine-tuned with 2K $k$-step UI Transition samples for each $k \in \{1, 2, 3, 4\}$, **without model-specific data filtering**. The target states $S_{t+k}$ with different $k$ values provide effective visual instructions for GUI agent training. TM: type match; GR: grounding accuracy for clicks; EM: exact match.

| Model | AndroidControl-Low | | | AndroidControl-High | | | GUI Odyssey | | |
|---|---|---|---|---|---|---|---|---|---|
| | TM | GR | EM | TM | GR | EM | TM | GR | EM |
| Qwen2.5-VL-7B | 94.9 | 90.9 | 83.8 | 72.9 | 66.6 | 59.2 | 59.8 | 47.5 | 44.9 |
| GUI-Shift-Qwen ($k=1$) | $98.0_{\uparrow 3.1}$ | $94.0_{\uparrow 3.1}$ | $90.6_{\uparrow 6.8}$ | $85.9_{\uparrow 13.0}$ | $77.5_{\uparrow 10.9}$ | $70.4_{\uparrow 11.2}$ | $78.5_{\uparrow 18.7}$ | $67.2_{\uparrow 19.7}$ | $54.8_{\uparrow 9.9}$ |
| GUI-Shift-Qwen ($k=2$) | $98.0_{\uparrow 3.1}$ | $93.7_{\uparrow 2.8}$ | $90.0_{\uparrow 6.2}$ | $85.9_{\uparrow 13.0}$ | $76.9_{\uparrow 10.3}$ | $70.8_{\uparrow 11.6}$ | $79.4_{\uparrow 19.6}$ | $68.6_{\uparrow 21.1}$ | $55.7_{\uparrow 10.8}$ |
| GUI-Shift-Qwen ($k=3$) | $97.9_{\uparrow 3.0}$ | $93.6_{\uparrow 2.7}$ | $91.5_{\uparrow 7.7}$ | $85.3_{\uparrow 12.4}$ | $77.7_{\uparrow 11.1}$ | $70.0_{\uparrow 10.8}$ | $77.1_{\uparrow 17.3}$ | $67.6_{\uparrow 20.1}$ | $53.8_{\uparrow 8.9}$ |
| GUI-Shift-Qwen ($k=4$) | $97.8_{\uparrow 2.9}$ | $92.8_{\uparrow 1.9}$ | $91.0_{\uparrow 7.2}$ | $84.8_{\uparrow 11.9}$ | $77.1_{\uparrow 10.5}$ | $69.6_{\uparrow 10.4}$ | $76.0_{\uparrow 16.2}$ | $65.7_{\uparrow 18.2}$ | $53.1_{\uparrow 8.2}$ |
| InternVL3-8B | 97.8 | 92.4 | 90.0 | 71.5 | 54.6 | 49.8 | 48.8 | 20.2 | 20.3 |
| GUI-Shift-Intern ($k=1$) | $98.1_{\uparrow 0.3}$ | $92.8_{\uparrow 0.4}$ | $89.6_{\downarrow 0.4}$ | $73.1_{\uparrow 1.6}$ | $56.0_{\uparrow 1.4}$ | $52.5_{\uparrow 2.7}$ | $42.9_{\downarrow 5.9}$ | $17.6_{\downarrow 2.6}$ | $17.9_{\downarrow 2.4}$ |
| GUI-Shift-Intern ($k=2$) | $97.6_{\downarrow 0.2}$ | $92.9_{\uparrow 0.5}$ | $87.9_{\downarrow 2.1}$ | $76.3_{\uparrow 4.8}$ | $59.8_{\uparrow 5.2}$ | $54.3_{\uparrow 4.5}$ | $59.4_{\uparrow 10.6}$ | $24.4_{\uparrow 4.2}$ | $23.1_{\uparrow 2.8}$ |
| GUI-Shift-Intern ($k=3$) | $97.5_{\downarrow 0.3}$ | $92.6_{\uparrow 0.2}$ | $89.3_{\downarrow 0.7}$ | $78.6_{\uparrow 7.1}$ | $61.7_{\uparrow 7.1}$ | $58.2_{\uparrow 8.4}$ | $51.4_{\uparrow 2.6}$ | $21.4_{\uparrow 1.2}$ | $21.4_{\uparrow 1.1}$ |
| GUI-Shift-Intern ($k=4$) | $97.3_{\downarrow 0.5}$ | $92.9_{\uparrow 0.5}$ | $88.0_{\downarrow 2.0}$ | $78.5_{\uparrow 7.0}$ | $64.3_{\uparrow 9.7}$ | $56.6_{\uparrow 6.8}$ | $59.6_{\uparrow 10.8}$ | $25.9_{\uparrow 5.7}$ | $23.3_{\uparrow 3.0}$ |
| Mimo-VL-7B-SFT | 90.8 | 93.5 | 85.7 | 75.2 | 75.7 | 63.1 | 86.9 | 66.3 | 62.0 |
| GUI-Shift-Mimo-SFT ($k=1$) | $98.6_{\uparrow 7.8}$ | $93.8_{\uparrow 0.3}$ | $92.0_{\uparrow 6.3}$ | $86.8_{\uparrow 11.6}$ | $74.3_{\downarrow 1.4}$ | $71.7_{\uparrow 8.6}$ | $85.4_{\downarrow 1.5}$ | $67.1_{\uparrow 0.8}$ | $61.3_{\downarrow 0.7}$ |
| GUI-Shift-Mimo-SFT ($k=2$) | $98.5_{\uparrow 7.7}$ | $92.7_{\downarrow 0.8}$ | $90.0_{\uparrow 4.3}$ | $87.0_{\uparrow 11.8}$ | $73.9_{\downarrow 1.8}$ | $70.3_{\uparrow 7.2}$ | $85.0_{\downarrow 1.9}$ | $66.0_{\downarrow 0.3}$ | $59.5_{\downarrow 2.5}$ |
| GUI-Shift-Mimo-SFT ($k=3$) | $98.2_{\uparrow 7.4}$ | $92.5_{\downarrow 1.0}$ | $88.4_{\uparrow 2.7}$ | $85.5_{\uparrow 10.3}$ | $72.2_{\downarrow 3.5}$ | $68.1_{\uparrow 5.0}$ | $86.0_{\downarrow 0.9}$ | $65.4_{\downarrow 0.9}$ | $58.5_{\downarrow 3.5}$ |
| GUI-Shift-Mimo-SFT ($k=4$) | $98.3_{\uparrow 7.5}$ | $93.0_{\downarrow 0.5}$ | $92.2_{\uparrow 6.5}$ | $86.9_{\uparrow 11.7}$ | $73.4_{\downarrow 2.3}$ | $72.4_{\uparrow 9.3}$ | $85.9_{\downarrow 1.0}$ | $67.7_{\uparrow 1.4}$ | $60.1_{\downarrow 1.9}$ |
| Mimo-VL-7B-RL | 91.8 | 94.5 | 87.2 | 76.5 | 77.5 | 64.6 | 87.2 | 67.9 | 63.1 |
| GUI-Shift-Mimo-RL ($k=1$) | $98.7_{\uparrow 6.9}$ | $94.7_{\uparrow 0.2}$ | $92.5_{\uparrow 5.3}$ | $86.7_{\uparrow 10.2}$ | $77.0_{\downarrow 0.5}$ | $70.8_{\uparrow 6.2}$ | $85.0_{\downarrow 2.2}$ | $67.8_{\downarrow 0.1}$ | $59.1_{\downarrow 4.0}$ |
| GUI-Shift-Mimo-RL ($k=2$) | $98.7_{\uparrow 6.9}$ | $93.3_{\downarrow 1.2}$ | $88.2_{\uparrow 1.0}$ | $86.1_{\uparrow 9.6}$ | $72.7_{\downarrow 4.8}$ | $69.0_{\uparrow 4.4}$ | $85.3_{\downarrow 1.9}$ | $66.2_{\downarrow 1.7}$ | $59.4_{\downarrow 3.7}$ |
| GUI-Shift-Mimo-RL ($k=3$) | $98.0_{\uparrow 6.2}$ | $93.8_{\downarrow 0.7}$ | $89.1_{\uparrow 1.9}$ | $85.8_{\uparrow 9.3}$ | $73.6_{\downarrow 3.9}$ | $68.8_{\uparrow 4.2}$ | $85.9_{\downarrow 1.3}$ | $66.0_{\downarrow 1.9}$ | $58.8_{\downarrow 4.3}$ |
| GUI-Shift-Mimo-RL ($k=4$) | $98.5_{\uparrow 6.7}$ | $93.0_{\downarrow 1.5}$ | $89.5_{\uparrow 2.3}$ | $86.6_{\uparrow 10.1}$ | $73.0_{\downarrow 4.5}$ | $70.1_{\uparrow 5.5}$ | $85.3_{\downarrow 1.9}$ | $66.1_{\downarrow 1.8}$ | $57.5_{\downarrow 5.6}$ |

Table 7: Performance on task automation benchmarks: AndroidControl and GUI Odyssey. Compared to training without data filtering, applying filtering yields greater improvements in most cases. TM: type match; GR: grounding accuracy for clicks; EM: exact match.

| Model | AndroidControl-Low | | | AndroidControl-High | | | GUI Odyssey | | |
|---|---|---|---|---|---|---|---|---|---|
| | TM | GR | EM | TM | GR | EM | TM | GR | EM |
| InternVL3-8B | 97.8 | 92.4 | 90.0 | 71.5 | 54.6 | 49.8 | 48.8 | 20.2 | 20.3 |
| GUI-Shift-Intern ($k=1$) | $96.2_{\downarrow1.6}$ | $92.6_{\uparrow0.2}$ | $87.0_{\downarrow3.0}$ | $78.6_{\uparrow7.1}$ | $63.2_{\uparrow8.6}$ | $55.1_{\uparrow5.3}$ | $51.6_{\uparrow2.8}$ | $24.5_{\uparrow4.3}$ | $22.1_{\uparrow1.8}$ |
| GUI-Shift-Intern ($k=2$) | $97.2_{\downarrow0.6}$ | $92.3_{\downarrow0.1}$ | $86.4_{\downarrow3.6}$ | $79.4_{\uparrow7.9}$ | $63.3_{\uparrow8.7}$ | $55.9_{\uparrow6.1}$ | $51.9_{\uparrow3.1}$ | $24.4_{\uparrow4.2}$ | $23.9_{\uparrow3.6}$ |
| GUI-Shift-Intern ($k=3$) | $96.3_{\downarrow1.5}$ | $92.7_{\uparrow0.3}$ | $87.7_{\downarrow2.3}$ | $79.9_{\uparrow8.4}$ | $65.5_{\uparrow10.9}$ | $57.6_{\uparrow7.8}$ | $59.5_{\uparrow10.7}$ | $27.2_{\uparrow7.0}$ | $24.8_{\uparrow4.5}$ |
| GUI-Shift-Intern ($k=4$) | $96.2_{\downarrow1.6}$ | $92.8_{\uparrow0.4}$ | $87.7_{\downarrow2.3}$ | $81.0_{\uparrow9.5}$ | $67.8_{\uparrow13.2}$ | $59.9_{\uparrow10.1}$ | $57.0_{\uparrow8.2}$ | $27.4_{\uparrow7.2}$ | $25.7_{\uparrow5.4}$ |
| Mimo-VL-7B-SFT | 90.8 | 93.5 | 85.7 | 75.2 | 75.7 | 63.1 | 86.9 | 66.3 | 62.0 |
| GUI-Shift-Mimo-SFT ($k=1$) | $97.8_{\uparrow7.0}$ | $93.8_{\uparrow0.3}$ | $92.2_{\uparrow6.5}$ | $84.6_{\uparrow9.4}$ | $75.7_{\uparrow0.0}$ | $70.2_{\uparrow7.1}$ | $81.8_{\downarrow5.1}$ | $64.3_{\downarrow2.0}$ | $56.2_{\downarrow5.8}$ |
| GUI-Shift-Mimo-SFT ($k=2$) | $98.6_{\uparrow7.8}$ | $93.3_{\downarrow0.2}$ | $92.9_{\uparrow7.2}$ | $86.4_{\uparrow11.2}$ | $75.8_{\uparrow0.1}$ | $72.0_{\uparrow8.9}$ | $85.2_{\downarrow1.7}$ | $66.6_{\uparrow0.3}$ | $60.3_{\downarrow1.7}$ |
| GUI-Shift-Mimo-SFT ($k=3$) | $98.6_{\uparrow7.8}$ | $94.0_{\uparrow0.5}$ | $93.2_{\uparrow7.5}$ | $87.2_{\uparrow12.0}$ | $75.6_{\downarrow0.1}$ | $73.4_{\uparrow10.3}$ | $86.1_{\downarrow0.8}$ | $66.3_{\uparrow0.0}$ | $60.7_{\downarrow1.3}$ |
| GUI-Shift-Mimo-SFT ($k=4$) | $98.6_{\uparrow7.8}$ | $93.5_{\uparrow0.0}$ | $92.7_{\uparrow7.0}$ | $85.8_{\uparrow10.6}$ | $73.9_{\downarrow1.8}$ | $71.6_{\uparrow8.5}$ | $85.9_{\downarrow1.0}$ | $67.5_{\uparrow1.2}$ | $60.9_{\downarrow1.1}$ |
| Mimo-VL-7B-RL | 91.8 | 94.5 | 87.2 | 76.5 | 77.5 | 64.6 | 87.2 | 67.9 | 63.1 |
| GUI-Shift-Mimo-RL ($k=1$) | $98.9_{\uparrow7.1}$ | $94.3_{\downarrow0.2}$ | $93.2_{\uparrow6.0}$ | $86.9_{\uparrow10.4}$ | $75.9_{\downarrow1.6}$ | $71.7_{\uparrow7.1}$ | $84.8_{\downarrow2.4}$ | $67.5_{\downarrow0.4}$ | $59.5_{\downarrow3.6}$ |
| GUI-Shift-Mimo-RL ($k=2$) | $97.7_{\uparrow5.9}$ | $93.7_{\downarrow0.8}$ | $91.3_{\uparrow4.1}$ | $87.6_{\uparrow11.1}$ | $75.9_{\downarrow1.6}$ | $71.6_{\uparrow7.0}$ | $84.9_{\downarrow2.3}$ | $65.4_{\downarrow2.5}$ | $58.9_{\downarrow4.2}$ |
| GUI-Shift-Mimo-RL ($k=3$) | $97.3_{\uparrow5.5}$ | $93.9_{\downarrow0.6}$ | $91.1_{\uparrow3.9}$ | $87.1_{\uparrow10.6}$ | $77.3_{\downarrow0.2}$ | $71.7_{\uparrow7.1}$ | $84.9_{\downarrow2.3}$ | $67.6_{\downarrow0.3}$ | $59.7_{\downarrow3.4}$ |
| GUI-Shift-Mimo-RL ($k=4$) | $96.8_{\uparrow5.0}$ | $94.2_{\downarrow0.3}$ | $90.9_{\uparrow3.7}$ | $87.0_{\uparrow10.5}$ | $77.0_{\downarrow0.5}$ | $72.1_{\uparrow7.5}$ | $84.7_{\downarrow2.5}$ | $68.3_{\uparrow0.4}$ | $59.9_{\downarrow3.2}$ |

# D  GROUNDING RESULTS WITH AND WITHOUT DATA FILTERING

We report the accuracy of GUI-Shift on two GUI grounding benchmarks: ScreenSpot-v2 and ScreenSpot-Pro, under both filtered and unfiltered settings.

For ScreenSpot-v2 (Table 8), all models except GUI-Shift-Intern achieve consistent improvements over their respective baselines without data filtering. With data filtering (Table 9), GUI-Shift-Intern models also surpass their baselines, indicating the benefit of filtering.

For ScreenSpot-Pro (Table 10), all GUI-Shift-Qwen models improve without data filtering, while other models exhibit mixed results across different $k$. We attribute the performance drop to the high-resolution desktop screenshots in ScreenSpot-Pro, which are not present in our training data. With data filtering (Table 11), GUI-Shift-Mimo-RL achieves consistent gains for all $k$.

Overall, GUI-Shift demonstrates strong generalization on GUI grounding, with performance gains in most cases, especially when data filtering is applied.

Table 8: Performance on GUI grounding benchmark: ScreenSpot-v2. Models are fine-tuned with 2K $k$-step UI Transition samples for each $k \in \{1, 2, 3, 4\}$, **without model-specific data filtering**.

| Model | Mobile | | | Desktop | | | Web | | | Avg. |
|---|---|---|---|---|---|---|---|---|---|---|
| | Text | Icon | Avg. | Text | Icon | Avg. | Text | Icon | Avg. | |
| Qwen2.5-VL-7B | 98.3 | 86.3 | 93.2 | 88.7 | 67.1 | 79.6 | 92.7 | 81.8 | 87.6 | 87.7 |
| GUI-Shift-Qwen ($k=1$) | 98.6 | 87.7 | 94.0 | 88.1 | 71.4 | 81.1 | 92.7 | 81.8 | 87.6 | 88.4$_{\uparrow 0.7}$ |
| GUI-Shift-Qwen ($k=2$) | 98.6 | 89.1 | 94.6 | 87.6 | 73.6 | 81.7 | 92.3 | 80.8 | 87.0 | 88.6$_{\uparrow 0.9}$ |
| GUI-Shift-Qwen ($k=3$) | 99.0 | 88.6 | 94.6 | 86.1 | 72.9 | 80.5 | 92.3 | 80.8 | 87.0 | 88.3$_{\uparrow 0.6}$ |
| GUI-Shift-Qwen ($k=4$) | 98.6 | 89.6 | 94.8 | 86.1 | 75.0 | 81.4 | 92.7 | 82.8 | 88.1 | 89.0$_{\uparrow 1.3}$ |
| InternVL3-8B | 93.4 | 81.5 | 88.4 | 80.4 | 52.1 | 68.6 | 91 | 73.4 | 82.8 | 81.3 |
| GUI-Shift-Intern ($k=1$) | 94.1 | 81.5 | 88.8 | 77.3 | 52.1 | 66.8 | 91.5 | 73.4 | 83.1 | 81.1$_{\downarrow 0.2}$ |
| GUI-Shift-Intern ($k=2$) | 93.8 | 80.6 | 88.2 | 80.4 | 52.1 | 68.6 | 91.5 | 72.4 | 82.6 | 81.1$_{\downarrow 0.2}$ |
| GUI-Shift-Intern ($k=3$) | 94.8 | 80.1 | 88.6 | 78.9 | 47.1 | 65.6 | 90.6 | 71.4 | 81.7 | 80.2$_{\downarrow 0.9}$ |
| GUI-Shift-Intern ($k=4$) | 94.1 | 82.0 | 89.0 | 79.4 | 54.3 | 68.9 | 91.9 | 71.9 | 82.6 | 81.5$_{\uparrow 0.2}$ |
| Mimo-VL-7B-SFT | 96.6 | 84.4 | 91.4 | 92.8 | 80.0 | 87.4 | 88.9 | 76.8 | 83.3 | 87.6 |
| GUI-Shift-Mimo-SFT ($k=1$) | 97.2 | 86.3 | 92.6 | 91.8 | 79.3 | 86.5 | 90.6 | 74.9 | 83.3 | 87.8$_{\uparrow 0.2}$ |
| GUI-Shift-Mimo-SFT ($k=2$) | 95.9 | 84.4 | 91.0 | 92.8 | 82.9 | 88.6 | 91.9 | 75.9 | 84.4 | 88.1$_{\uparrow 0.5}$ |
| GUI-Shift-Mimo-SFT ($k=3$) | 96.6 | 86.3 | 92.2 | 91.8 | 82.9 | 88.0 | 90.6 | 77.3 | 84.4 | 88.4$_{\uparrow 0.8}$ |
| GUI-Shift-Mimo-SFT ($k=4$) | 96.9 | 87.2 | 92.8 | 91.8 | 84.3 | 88.6 | 89.7 | 80.3 | 85.4 | 89.2$_{\uparrow 1.6}$ |
| Mimo-VL-7B-RL | 98.3 | 86.3 | 93.2 | 90.2 | 80.7 | 86.2 | 92.7 | 75.4 | 84.7 | 88.4 |
| GUI-Shift-Mimo-RL ($k=1$) | 99.3 | 88.2 | 94.6 | 91.8 | 80.7 | 87.1 | 90.2 | 75.4 | 83.3 | 88.8$_{\uparrow 0.4}$ |
| GUI-Shift-Mimo-RL ($k=2$) | 97.9 | 87.2 | 93.4 | 91.8 | 81.4 | 87.4 | 91 | 74.4 | 83.3 | 88.4$_{\uparrow 0.0}$ |
| GUI-Shift-Mimo-RL ($k=3$) | 98.3 | 87.2 | 93.6 | 91.8 | 83.6 | 88.3 | 91 | 76.4 | 84.2 | 89.0$_{\uparrow 0.6}$ |
| GUI-Shift-Mimo-RL ($k=4$) | 99.3 | 87.7 | 94.4 | 91.8 | 81.4 | 87.4 | 90.6 | 75.4 | 83.5 | 88.8$_{\uparrow 0.4}$ |

Table 9: Performance on GUI grounding benchmark: ScreenSpot-v2. **Model-specific data filtering is applied** to InternVL3-8B, MimoVL-7B-SFT, and MimoVL-7B-RL. For each model, we select 2K $k$-step UI Transition samples for each $k \in \{1, 2, 3, 4\}$ from a pool of candidates.

| Model | Mobile | | | Desktop | | | Web | | | Avg. |
|---|---|---|---|---|---|---|---|---|---|---|
| | Text | Icon | Avg. | Text | Icon | Avg. | Text | Icon | Avg. | |
| InternVL3-8B | 93.4 | 81.5 | 88.4 | 80.4 | 52.1 | 68.6 | 91.0 | 73.4 | 82.8 | 81.3 |
| GUI-Shift-Intern ($k=1$) | 93.8 | 83.4 | 89.4 | 80.4 | 51.4 | 68.3 | 91.0 | 73.4 | 82.8 | 81.6$\uparrow$0.3 |
| GUI-Shift-Intern ($k=2$) | 94.5 | 82 | 89.2 | 78.9 | 51.4 | 67.4 | 91.9 | 72.4 | 82.8 | 81.3$\uparrow$0.0 |
| GUI-Shift-Intern ($k=3$) | 94.5 | 83.4 | 89.8 | 79.4 | 50.7 | 67.4 | 91.9 | 73.4 | 83.3 | 81.7$\uparrow$0.4 |
| GUI-Shift-Intern ($k=4$) | 93.4 | 83.4 | 89.2 | 77.8 | 54.3 | 68.0 | 91.5 | 73.9 | 83.3 | 81.6$\uparrow$0.3 |
| Mimo-VL-7B-SFT | 96.6 | 84.4 | 91.4 | 92.8 | 80.0 | 87.4 | 88.9 | 76.8 | 83.3 | 87.6 |
| GUI-Shift-Mimo-SFT ($k=1$) | 98.3 | 87.7 | 93.8 | 92.3 | 82.1 | 88.0 | 94.0 | 79.8 | 87.4 | 90.1$\uparrow$2.5 |
| GUI-Shift-Mimo-SFT ($k=2$) | 98.6 | 87.7 | 94.0 | 93.3 | 83.6 | 89.2 | 92.7 | 79.8 | 86.7 | 90.3$\uparrow$2.7 |
| GUI-Shift-Mimo-SFT ($k=3$) | 96.6 | 85.3 | 91.8 | 91.8 | 81.4 | 87.4 | 89.3 | 80.3 | 85.1 | 88.4$\uparrow$0.8 |
| GUI-Shift-Mimo-SFT ($k=4$) | 97.6 | 83.9 | 91.8 | 92.3 | 82.9 | 88.3 | 91.5 | 79.3 | 85.8 | 88.8$\uparrow$1.2 |
| Mimo-VL-7B-RL | 98.3 | 86.3 | 93.2 | 90.2 | 80.7 | 86.2 | 92.7 | 75.4 | 84.7 | 88.4 |
| GUI-Shift-Mimo-RL ($k=1$) | 99.0 | 87.7 | 94.2 | 91.2 | 83.6 | 88.0 | 89.7 | 72.9 | 81.9 | 88.4$\uparrow$0.0 |
| GUI-Shift-Mimo-RL ($k=2$) | 97.9 | 86.7 | 93.2 | 90.7 | 80.7 | 86.5 | 91.0 | 75.4 | 83.8 | 88.2$\downarrow$0.2 |
| GUI-Shift-Mimo-RL ($k=3$) | 99.0 | 87.2 | 94.0 | 91.8 | 80.7 | 87.1 | 91.0 | 76.8 | 84.4 | 88.9$\uparrow$0.5 |
| GUI-Shift-Mimo-RL ($k=4$) | 98.6 | 85.3 | 93 | 91.8 | 81.4 | 87.4 | 91.9 | 77.3 | 85.1 | 88.8$\uparrow$0.4 |

Table 10: Performance on GUI grounding benchmark: ScreenSpot-Pro. Models are fine-tuned with 2K $k$-step UI Transition samples for each $k \in \{1, 2, 3, 4\}$, **without model-specific data filtering**.

| Model | CAD | | Dev | | Creative | | Scientific | | Office | | OS | | Avg. |
|---|---|---|---|---|---|---|---|---|---|---|---|---|---|
| | Text | Icon | Text | Icon | Text | Icon | Text | Icon | Text | Icon | Text | Icon | |
| Qwen2.5-VL-7B | 16.2 | 1.6 | 44.2 | 2.1 | 36.9 | 7.7 | 47.9 | 8.2 | 53.7 | 18.9 | 36.4 | 7.9 | 26.4 |
| GUI-Shift-Qwen ($k=1$) | 16.2 | 4.7 | 52.6 | 9.0 | 27.3 | 7.0 | 52.1 | 5.5 | 49.7 | 17.0 | 38.3 | 13.5 | 26.8$\uparrow$0.4 |
| GUI-Shift-Qwen ($k=2$) | 16.8 | 3.1 | 52.6 | 10.3 | 29.3 | 8.4 | 52.1 | 4.5 | 50.8 | 17.0 | 35.5 | 13.5 | 27.2$\uparrow$0.8 |
| GUI-Shift-Qwen ($k=3$) | 15.7 | 3.1 | 52.6 | 9.7 | 30.3 | 7.0 | 50.7 | 5.5 | 48.0 | 15.1 | 35.5 | 12.4 | 26.5$\uparrow$0.1 |
| GUI-Shift-Qwen ($k=4$) | 17.3 | 3.1 | 51.9 | 9.7 | 30.3 | 7.0 | 54.2 | 5.5 | 49.7 | 17.0 | 33.6 | 12.4 | 27.1$\uparrow$0.7 |
| InternVL3-8B | 8.6 | 4.7 | 27.3 | 4.1 | 27.3 | 0.7 | 24.3 | 4.5 | 32.2 | 3.8 | 11.2 | 3.4 | 15.0 |
| GUI-Shift-Intern ($k=1$) | 10.2 | 1.6 | 27.3 | 4.8 | 26.8 | 0.7 | 23.6 | 3.6 | 33.9 | 7.5 | 15.0 | 2.2 | 15.4$\uparrow$0.4 |
| GUI-Shift-Intern ($k=2$) | 7.6 | 3.1 | 27.3 | 3.4 | 28.3 | 0.7 | 21.5 | 4.5 | 33.3 | 7.5 | 12.1 | 2.2 | 14.9$\downarrow$0.1 |
| GUI-Shift-Intern ($k=3$) | 9.6 | 3.1 | 23.4 | 5.5 | 26.3 | 0.7 | 20.1 | 1.8 | 29.9 | 3.8 | 11.2 | 1.1 | 13.7$\downarrow$1.3 |
| GUI-Shift-Intern ($k=4$) | 9.1 | 4.7 | 24.0 | 4.1 | 28.8 | 0.7 | 21.5 | 3.6 | 31.1 | 3.8 | 13.1 | 1.1 | 14.5$\downarrow$0.5 |
| Mimo-VL-7B-SFT | 47.2 | 23.4 | 48.7 | 9.0 | 48.0 | 13.3 | 70.8 | 27.3 | 64.4 | 39.6 | 36.4 | 15.7 | 39.8 |
| GUI-Shift-Mimo-SFT ($k=1$) | 49.7 | 15.6 | 46.8 | 13.8 | 49.0 | 16.1 | 74.3 | 25.5 | 65.0 | 37.7 | 40.2 | 15.7 | 40.9$\uparrow$1.1 |
| GUI-Shift-Mimo-SFT ($k=2$) | 48.7 | 20.3 | 50.6 | 11.7 | 48.5 | 14.0 | 72.9 | 26.4 | 63.3 | 41.5 | 36.4 | 16.9 | 40.6$\uparrow$0.8 |
| GUI-Shift-Mimo-SFT ($k=3$) | 45.7 | 18.8 | 48.1 | 12.4 | 45.5 | 12.6 | 70.1 | 28.2 | 65.5 | 39.6 | 35.5 | 19.1 | 39.6$\downarrow$0.2 |
| GUI-Shift-Mimo-SFT ($k=4$) | 45.7 | 18.8 | 51.3 | 12.4 | 46.5 | 12.6 | 71.5 | 28.2 | 62.7 | 41.5 | 37.4 | 16.9 | 39.9$\uparrow$0.1 |
| Mimo-VL-7B-RL | 48.2 | 14.1 | 46.8 | 11.0 | 47.0 | 14.0 | 71.5 | 27.3 | 66.7 | 39.6 | 39.3 | 19.1 | 40.2 |
| GUI-Shift-Mimo-RL ($k=1$) | 48.7 | 15.6 | 46.1 | 13.8 | 51.0 | 13.3 | 72.2 | 30.0 | 66.7 | 43.4 | 42.1 | 21.3 | 41.7$\uparrow$1.5 |
| GUI-Shift-Mimo-RL ($k=2$) | 46.7 | 14.1 | 47.4 | 13.8 | 49.0 | 12.6 | 70.1 | 27.3 | 65.5 | 39.6 | 41.1 | 21.3 | 40.5$\uparrow$0.3 |
| GUI-Shift-Mimo-RL ($k=3$) | 49.7 | 12.5 | 47.4 | 11.0 | 47.0 | 11.2 | 70.8 | 26.4 | 65.5 | 45.3 | 35.5 | 20.2 | 39.9$\downarrow$0.3 |
| GUI-Shift-Mimo-RL ($k=4$) | 46.7 | 14.1 | 46.1 | 13.8 | 47.0 | 14.7 | 69.4 | 27.3 | 64.4 | 41.5 | 36.4 | 21.3 | 39.8$\downarrow$0.4 |

Table 11: Performance on GUI grounding benchmark: ScreenSpot-Pro. **Model-specific data filtering is applied** to InternVL3-8B, MimoVL-7B-SFT, and MimoVL-7B-RL. For each model, we select 2K $k$-step UI Transition samples for each $k \in \{1, 2, 3, 4\}$ from a pool of candidates.

| Model | CAD | | Dev | | Creative | | Scientific | | Office | | OS | | Avg. |
|---|---|---|---|---|---|---|---|---|---|---|---|---|---|
| | Text | Icon | Text | Icon | Text | Icon | Text | Icon | Text | Icon | Text | Icon | |
| InternVL3-8B | 8.6 | 4.7 | 27.3 | 4.1 | 27.3 | 0.7 | 24.3 | 4.5 | 32.2 | 3.8 | 11.2 | 3.4 | 15.0 |
| GUI-Shift-Intern ($k=1$) | 11.2 | 3.1 | 26.6 | 5.5 | 28.8 | 0.7 | 20.1 | 4.5 | 33.9 | 7.5 | 13.1 | 1.1 | 15.4↑0.4 |
| GUI-Shift-Intern ($k=2$) | 8.1 | 1.6 | 26.0 | 4.1 | 26.3 | 0.7 | 25.7 | 1.8 | 31.1 | 5.7 | 14.0 | 1.1 | 14.5↓0.5 |
| GUI-Shift-Intern ($k=3$) | 8.6 | 1.6 | 27.3 | 4.1 | 25.3 | 0.7 | 21.5 | 1.8 | 33.3 | 5.7 | 14.0 | 1.1 | 14.4↓0.6 |
| GUI-Shift-Intern ($k=4$) | 8.1 | 1.6 | 27.9 | 4.8 | 26.8 | 0.0 | 22.9 | 2.7 | 32.8 | 3.8 | 10.3 | 2.2 | 14.5↓0.5 |
| Mimo-VL-7B-SFT | 47.2 | 23.4 | 48.7 | 9.0 | 48.0 | 13.3 | 70.8 | 27.3 | 64.4 | 39.6 | 36.4 | 15.7 | 39.8 |
| GUI-Shift-Mimo-SFT ($k=1$) | 49.2 | 14.1 | 50.0 | 9.0 | 48.0 | 11.9 | 73.6 | 29.1 | 54.3 | 43.4 | 38.3 | 15.7 | 40.7↑0.9 |
| GUI-Shift-Mimo-SFT ($k=2$) | 46.2 | 17.2 | 46.8 | 11.0 | 47.5 | 10.5 | 74.3 | 30.9 | 65.5 | 43.4 | 35.5 | 16.9 | 40.0↑0.2 |
| GUI-Shift-Mimo-SFT ($k=3$) | 50.8 | 15.6 | 47.4 | 13.8 | 47.0 | 12.6 | 69.4 | 24.5 | 65.0 | 41.5 | 38.3 | 19.1 | 40.2↑0.4 |
| GUI-Shift-Mimo-SFT ($k=4$) | 44.2 | 18.8 | 53.9 | 11.0 | 43.9 | 11.2 | 72.2 | 26.4 | 62.1 | 47.2 | 40.2 | 12.4 | 39.4↓0.4 |
| Mimo-VL-7B-RL | 48.2 | 14.1 | 46.8 | 11.0 | 47.0 | 14.0 | 71.5 | 27.3 | 66.7 | 39.6 | 39.3 | 19.1 | 40.2 |
| GUI-Shift-Mimo-RL ($k=1$) | 48.7 | 14.1 | 51.9 | 13.1 | 50.0 | 15.4 | 70.8 | 28.2 | 67.2 | 39.6 | 37.4 | 21.3 | 41.6↑1.4 |
| GUI-Shift-Mimo-RL ($k=2$) | 47.2 | 18.8 | 47.4 | 14.5 | 49.0 | 15.4 | 72.2 | 29.1 | 66.1 | 43.4 | 39.3 | 19.1 | 41.3↑0.9 |
| GUI-Shift-Mimo-RL ($k=3$) | 45.7 | 17.2 | 46.1 | 11.7 | 49.5 | 14.0 | 70.8 | 29.1 | 66.1 | 39.6 | 37.4 | 21.3 | 40.4↑0.2 |
| GUI-Shift-Mimo-RL ($k=4$) | 47.7 | 18.8 | 48.1 | 13.8 | 52.5 | 12.6 | 72.2 | 28.2 | 67.2 | 45.3 | 40.2 | 20.2 | 41.8↑1.6 |

# E  REWARD DISTRIBUTION OF MODEL-SPECIFIC DATA FILTERING

To examine the relationship between the $K$ value, task difficulty, and model capability, we analyze the reward distributions obtained during the data filtering stage. Our analysis is based on 7,783 shared samples. For fair comparison, we use exactly the same samples for each $K$ across all models, and the current state $S(t)$ and ground-truth actions remain identical across different $K$ values. Figure 5 presents the reward distributions for each model across different $K$ values.

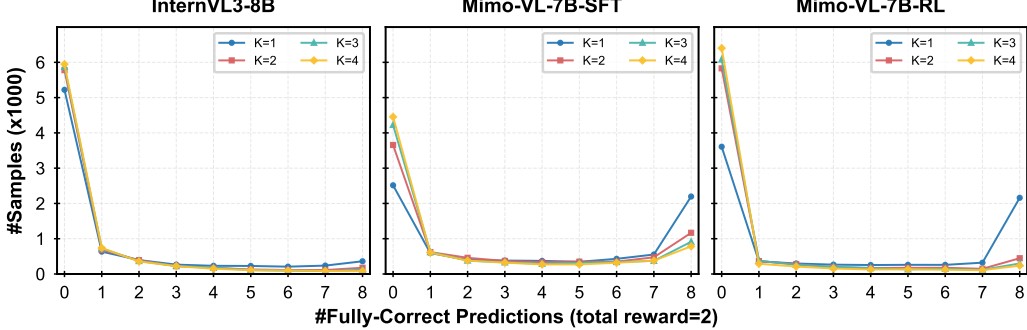

Figure 5: Comparison of reward distributons during model-specific data filtering. The x-axis represents the number of fully-correct predictions, where $x = 0$ indicates all 8 rollouts are incorrect and $x = 8$ indicates all 8 rollouts are correct. The y-axis reports how many K-step GUI Transition samples fall into each fully-correct count.

We provide detailed analysis from two perspectives:

• **From the perspective of $K$.** (1) At $x = 0$, where all 8 rollouts are incorrect, the sample counts follow $K = 4 > 3 > 2 > 1$, indicating that larger $K$ values correspond to higher difficulty. (2) At $x = 8$, where all 8 predictions are correct, the trend reverses to $K = 1 > 2 > 3 > 4$, showing that smaller $K$ values are easier for the models.

• **From the perspective of model capability.** As shown in Table 1 and Table 2, Mimo-VL-7B-SFT and Mimo-VL-7B-RL perform better than InternVL3-8B on four GUI-related benchmarks. We may

reasonably consider the Mimo models to have stronger GUI capability. A similar pattern appears in Figure 5, both Mimo models have noticeably more samples at $x = 8$ than InternVL3-8B, suggesting that models with stronger GUI capability perform better on the K-step GUI Transition task.

Overall, these findings reflects that the K-step GUI Transition task becomes more challenging as $K$ increases, and that models with stronger GUI capability achieve better performance on this task, which aligns with intuition.

# F    INTERNVL3-8B PERFORMANCE SCALING WITH TRAINING DATA SIZE

To investigate how training data size affects GUI-Shift performance, we scale the training set from 2K to 6K samples using $K$-step GUI Transition data ($K = 1$). Figure 6 illustrates the results on AndroidControl-Low and AndroidControl-High.

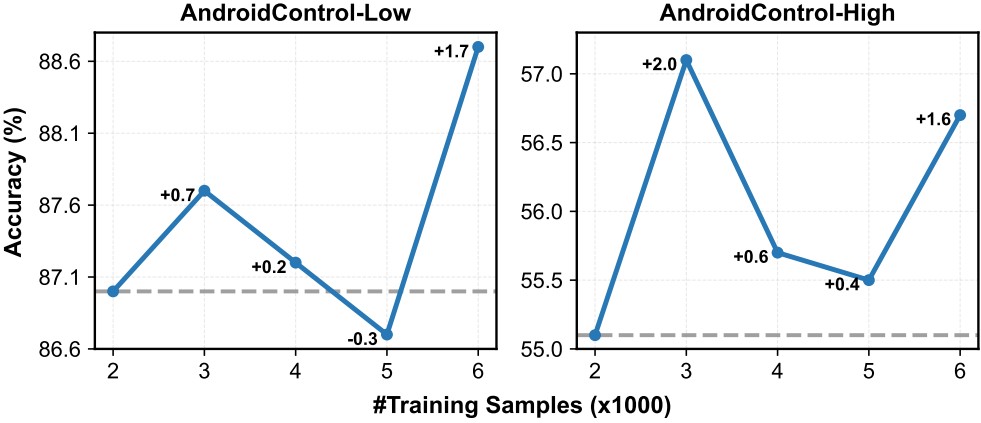

Figure 6: Performance on AndroidControl with InternVL3-8B trained on 1-step GUI Transition data. Training samples are scaled from 2K to 6K. Accuracy exhibits an overall upward trend with only minor fluctuations.

Specifically, relative to the 2K setting, GUI-Shift-Intern shows an additional gain of 1.7% when trained with 6K samples on AndroidControl-Low, and a further 2.0% improvement when trained with 3K samples on AndroidControl-High. In general, we observe an overall upward trend in accuracy as the data size increases, with only small fluctuations across different scales.

