# OpenReview forum: "GUI-Shift: Enhancing VLM-Based GUI Agents through Self-supervised Reinforcement Learning"
_ICLR.cc/2026/Conference — ICLR 2026 Poster_

### Official Review · Reviewer_hoLT · 2025-10-29

**Soundness:** 3
**Presentation:** 3
**Contribution:** 3
**Rating:** 6
**Confidence:** 3

**Summary:**

The paper introduces a novel self-supervised task for VLMs used in GUI agents. It is inspired by inverse dynamics within unlabeled GUI trajectories. They also propose a self-supervised RL framework with advantages in handling action multiplicity and poor generalization in GUI tasks.

**Strengths:**

- The primary strength is the "K-step GUI Transition" task. By formulating the training problem as a self-supervised inverse dynamics task , the authors provide a solution to bypass the need for expensive natural language annotations. This addresses the scalability bottleneck in GUI agent training data.
- The paper provides a well-crafted design of a RL-framework. Novelly set the visual goal as the supervision signal. This design requires the VLMs to capture both the diverse GUI settings and the dynamics transformation between states, therefore enhancing the situational awareness of the model.
- The experimental quality is a major strength. The authors validate their method across multiple VLM backbones and diverse benchmarks (automation and grounding). The extensive ablation studies (covering filtering , task formulation , training algorithm , and reasoning ) are thorough and leave little doubt about the efficacy of each component.
- The paper is exceptionally clear. Figure 1  is a model for how to visually communicate a complex framework, perfectly contrasting the proposed method with prior work

**Weaknesses:**

See Questions

**Questions:**

• Formatting Suggestion: A minor formatting note: On page 4, Line 254, the label “(3)” appears misaligned.

• Predicting Action Sequences: The current K-step GUI Transition task only requires the model to predict the first action initiating the transition from $S_t$ to $S_{t+k}$. However, the "Action Prediction" example in Figure 1 (Middle) implies the model reasons about a sequence of actions ("...I should click the search box and type... So, I think the first step should be: click..."). Have the authors considered or experimented with a more complex task objective, such as predicting the full sequence of $k$ actions required to reach $S_{t+k}$? If so, how did this compare to predicting only the initial action?

• Generality of Data Filtering: The data filtering pipeline, which selects samples with both correct and incorrect predictions, is presented as a key component for improving performance (Figure 2). However, this filtering was not applied to the Qwen2.5-VL-7B model because it produced an "exceptionally high" proportion of all-correct or all-incorrect samples5. This suggests the filtering heuristic may be model-dependent or brittle. Could the authors comment on the generality of this filtering component? Does this finding imply that the heuristic's effectiveness is strongly tied to the base model's initial capabilities or calibration?

• Impact of Omitting Reasoning Traces: The ablation study in Table 3 shows that omitting explicit reasoning traces during training not only nearly halves training time but also improves downstream performance. This finding is significant and somewhat counter-intuitive. Could the authors provide further explanation or hypotheses for this phenomenon?

• Analysis of the $k$ Parameter: The choice of $k$ in the K-step GUI Transition task is a central hyperparameter. Based on Tables 1 and 2, the optimal $k$ varies by model and task. For instance, $k=1$ (predicting the transition to the immediate next state) often performs very well, particularly for GUI grounding (best for 3/4 models). Conversely, Qwen2.5-VL-7B and InternVL3-8B achieved their best automation scores with $k=4$. Do the authors have any insights into this behavior? Is there an observable trade-off between the signal clarity of small $k$ values and the long-term goal-conditioning provided by larger $k$ values?

---

> ### Author Response · Authors · 2025-11-20
> **Response to Reviewer hoLT (Part 1/2)**
>
> We sincerely thank the reviewer for the kind words and appreciate that you recognize the novelty and significance of our work. We treasure the opportunity to address your concerns.
>
> ---
>
> > Question 1: A minor formatting note: On page 4, Line 254
>
> We appreciate the careful reading. We will optimize it in the next version.
>
> ---
>
> >  Question 2: How about predicting the full sequence of actions required to reach? If so, how did this compare to predicting only the initial action?
>
> Thank you for the insightful question. We have carefully considered the possibility of predicting multi-step or full action sequences, but we have several concerns about this approach.
>
> The K-step GUI Transition task is intentionally designed to let the model compare the current state $S(t)$ with the future state $S(t+k)$ and decide the action that should be taken at the current state. This setup is also consistent with real interaction settings, where decisions are always made based on the current screen.
>
> In contrast, predicting a multi-step sequence would require the model to output later actions without access to the intermediate screens. After the first action, the model would no longer see the updated state, making key information such as click locations difficult to infer and difficult to control.
>
> If you have any further questions, please don’t hesitate to let us know. We would be glad to assist in any way we can.
>
> ---
>
> >  Question 3: What is the generality of the model-specific data filtering mechanism, and is its effectiveness strongly tied to the base model’s initial capabilities or calibration?
>
> Thank you for raising this concern. To better illustrate the behavior of the filtering mechanism on Qwen2.5-VL-7B, we report the distribution of samples according to **how many of the 8 rollouts produce the correct action**, using the same data filtering setup as for the other models. The table below summarizes the results.
>
> | K   | Total Samples | All Incorrect (0) | Valid filtered (1-7) | All Correct (8) |
> | --- | ------------- | ----------------- | -------------------- | --------------- |
> | 1   | 41,614        | 36,219            | 381                  | 5,014           |
> | 2   | 38,500        | 36,866            | 240                  | 1,394           |
> | 3   | 3,900         | 3,769             | 27                   | 104             |
> | 4   | 4,700         | 4,576             | 24                   | 100             |
>
> These results show that the distribution is dominated by samples where "all 8 rollouts are correct" or "all 8 rollouts are incorrect". This pattern reflects that under a high temperature, Qwen2.5-VL-7B produces highly consistent outputs. As a result, obtaining a balanced set of "1-7 correct" samples requires sampling more trajectories, which increases the computational cost but remains feasible.
>
> For models that initially perform poorly on the K-step GUI Transition task and produce too few valid filtered samples (e.g., when most predictions are incorrect), one option is to apply a small amount of SFT on K-step GUI Transition data first to improve their basic capability and then perform model-specific filtering followed by RL training.
>
> Overall, we believe this filtering process is practical in most cases and can adapt to models with different levels of initial capability.
>
> If you have any other questions or need further clarification on the details, please feel free to let us know. We’d be happy to assist you.

---

> ### Author Response · Authors · 2025-11-20
> **Response to Reviewer hoLT (Part 2/2)**
>
> > Question 4: Why does omitting explicit reasoning traces during training lead to improved downstream performance?
>
> Thank you for the question. Our decision to omit reasoning during training is based on our empirical findings that it does not improve performance on the tested mobile GUI tasks.
>
> In our early experiments, we asked the model to produce its own reasoning traces. Since our method does not include an initial stage that teaches the model how to reason, we were concerned that a 7B model might generate unreliable reasoning, which could introduce noise into the training process. Based on this concern, we tried removing reasoning entirely. As shown in Table 3, the model performs better without explicit reasoning, and we therefore adopted this setting for all large-scale experiments. A possible explanation is that for smaller models and for our single-step prediction task, additional reasoning may cause instability instead of helping.
>
> Please don’t hesitate to reach out with any additional questions or comments. I would be more than happy to assist further.
>
> ---
>
> > Question 5: Why does the optimal value of $k$ vary across models and tasks, and is there a trade-off between smaller and larger $k$?
>
> Thank you for the thoughtful question. We would first like to clarify that the models shown in Tables 1 and 2 are not the best-performing variants for every individual benchmark. Instead, we select the variants that exhibit the most balanced performance across benchmarks. The full results for all model variants, including both with and without data filtering, are provided in Tables 5-9 in Appendix B and Appendix C.
>
> Since the model-specific data filtering mechanism yields different filtered datasets for different models, it is difficult to isolate the effect of $k$ when looking only at the filtered versions. To better isolate the effect of $k$, we would like to point to the results **without** data filtering, shown as the red curves in Figure 2. In this setting, the variables are carefully controlled:
> - For each model, the training samples for different values of $k$ share the same current state $S(t)$ and the same ground-truth action.
> - Across different models, the training data for the same $k$ is identical.
> This allows us to focus purely on how $k$ affects performance. From Figure 2(a), (f), and (k), we observe that on AndroidControl-Low, $k=1$ tends to give slightly better results.  From Figure 2(b), (g), and (l),  on AndroidControl-High, $k=4$ performs slightly better. This pattern is consistent with the task definitions: AndroidControl-Low is a short-horizon setting where the next action is directly guided by step instructions, making small $k$ more suitable; while AndroidControl-High requires following task-level instructions, which benefits more from the larger $k$. We think these observations align with our intuition.
>
> If you have any additional questions about the details, please don't hesitate to reach out. We would be glad to provide further clarification.
>
> ---
>
> Thank you once again for your positive feedback! We hope our responses have addressed your questions, and we are very happy to provide any further clarification if needed.

---

### Official Review · Reviewer_7wjQ · 2025-10-31

**Soundness:** 3
**Presentation:** 3
**Contribution:** 3
**Rating:** 6
**Confidence:** 4

**Summary:**

The paper proposes GUI-Shift, a self-supervised reinforcement learning framework for training VLM-based GUI agents without manual task instructions. The core pretext task is K-step GUI Transition: given a current screenshot and a target future screenshot, the model predicts the first action that initiates the transition. Training uses GRPO with a simple, verifiable reward, and a model-specific data filtering step that retains samples where the model produces both correct and incorrect actions under stochastic decoding. The authors fine-tune four open VLMs and report gains over base models on AndroidControl, GUI Odyssey, ScreenSpot-v2, and ScreenSpot-Pro.

**Strengths:**

S1: The K-step Transition objective is a compact way to leverage unlabeled trajectories at scale. The visual-goal formulation reduces reliance on noisy textual instructions and naturally encourages temporal reasoning.


S2: GRPO avoids penalizing multiple valid clicks (any point in a control) and removes the need for a critic. The binary-format action reward is stable, inexpensive, and reproducible.

S3: On AndroidControl-High, GUI-Shift-Qwen improves EM by +11.2% (to 70.4%), with other base models also benefiting. Despite training only on AndroidControl trajectories, models improve on ScreenSpot-v2 and ScreenSpot-Pro, demonstrating potential cross-platform generalization for GUI grounding.

**Weaknesses:**

W1: Training relies solely on the AndroidControl split, which undercuts the claim that the method scales to arbitrary “unlabeled trajectories.”

W2: Performance dips on GUI Odyssey for some models are attributed to tablet layouts; ScreenSpot‑Pro (desktop, high‑res) gains are modest.

W3: The work evaluates offline action accuracy and grounding. Demonstrating end‑to‑end success in a dynamic environment like AndroidWorld would strengthen claims about task automation robustness.

**Questions:**

(1) What fraction of candidate pairs survive filtering per model and per k? What is the inter-model overlap of filtered pairs?

(2) In Table 3, were the task/step instruction baselines also trained with GRPO and the same binary-format reward?

(3) If only click points are recorded, how is the box inferred, and how would GUI‑Shift operate on real logs that lack element metadata?

---

> ### Author Response · Authors · 2025-11-20
> **Response to Reviewer 7wjQ (Part 1/2)**
>
> We appreciate that the reviewer recognizes the significance and effectiveness of our work. We treasure the opportunity to address your concerns.
>
> ---
>
> >  Weakness 1: Training relies solely on the AndroidControl split.
>
> Thank you for raising this concern. One reason we use AndroidControl is that it allows a fair comparison with training setups based on task instructions and step instructions, as reported in Table 3. We would also like to clarify that our method relies only on the raw trajectories and does not use any annotations during training. We plan to extend our experiments to other mobile GUI datasets in future work to further show the applicability of our approach.
>
> If you have any other questions or would like further clarification, please feel free to let us know. We would be happy to assist.
>
> ---
>
> >  Weakness 2: The improvements on GUI Odyssey (which includes tablet layouts) and on ScreenSpot-Pro (desktop and high-resolution GUIs) are modest.
>
> Thank you for raising this concern. As shown in Table 2, all four models show consistent gains on ScreenSpot-Pro. In Table 5 of Appendix C, Qwen2.5-VL-7B improves by 10.8% on GUI Odyssey, and in Table 6, InternVL3-8B improves by 5.4%. We acknowledge that the improvements on these two benchmarks may be incremental compared with some GUI-specific VLMs, but our models achieve clear gains on AndroidControl, reaching up to 11.6% on AndroidControl-High for GUI-Shift-Qwen (k=2), as shown in Table 5. Given that our training data contains only 2K mobile GUI samples, we believe these improvements are reasonable and acceptable.
>
> In addition, we believe that GUI-Shift can also be applied to other platforms such as web or desktop GUIs, since the same K-step GUI Transition task can be constructed directly from unlabeled trajectories collected on those platforms. We will continue to explore these directions as part of our future work.
>
> If you have any further questions, please feel free to ask. We would be happy to assist.
>
> ---
>
> > Weakness 3: Evaluation results of end‑to‑end success rate.
>
> Thank you for this concern. We now provide the end-to-end success rate on AndroidControl and AndroidWorld [1]. Please refer to General Response 1 for more details. We are actively conducting more experiments on interactive benchmarks with more models.
>
> [1] Rawles, Christopher, et al. "Androidworld: A dynamic benchmarking environment for autonomous agents." _arXiv preprint arXiv: 2405.14573_ (2024).
>
> ---
>
> >  Question 1.1: What fraction of candidate pairs survive filtering per model and per k?
>
> Thank you for raising this question. To report the fraction of candidate pairs that survive filtering for each model and each $K$, we identify 7,783 shared samples across all models. These are samples for which every model produced rollouts, allowing us to analyze filtering behavior under exactly the same data. For fair comparison, each model uses the same samples for every $K$, and the current state $S(t)$ and ground-truth actions remain identical across different $K$ values. This ensures that the filtering outcomes are directly comparable.
>
> The table below shows the proportion of samples with 1-7 correct predictions (i.e., those that pass filtering):
>
> | Model          | K=1           | K=2           | K=3           | K=4           |
> | -------------- | ------------- | ------------- | ------------- | ------------- |
> | InternVL3-8B   | 2,201 (28.3%) | 1,828 (23.5%) | 1,789 (23.0%) | 1,754 (22.5%) |
> | Mimo-VL-7B-SFT | 3,072 (39.5%) | 2,958 (38.0%) | 2,645 (34.0%) | 2,545 (32.7%) |
> | Mimo-VL-7B-RL  | 2,015 (25.9%) | 1,507 (19.4%) | 1,406 (18.1%) | 1,143 (14.7%) |
>
> We further analyze the full reward distributions in **Appendix E**, which provides additional insight into how $K$ and model capability influence filtering outcomes. If the reviewer is interested in a more detailed view, we kindly invite you to refer to that section.
>
> If you have any additional questions about the details, please don't hesitate to reach out. We would be glad to provide further clarification.

---

> ### Author Response · Authors · 2025-11-20
> **Response to Reviewer 7wjQ (Part 2/2)**
>
> >  Question 1.2: What is the inter-model overlap of filtered pairs?
>
> Thank you for the question. To quantify the inter-model overlap of filtered pairs, we compute how many samples appear in the filtered sets of one, two, or all three models. For each $K$, the three models together contribute 6,000 filtered samples (as we use 2,000 samples to train each model). After deduplication, the number of distinct samples and their overlap are shown below:
>
> | K   | Distinct Samples | In 1 Model | In 2 Model | In 3 Model |
> | --- | ---------------- | ---------- | ---------- | ---------- |
> | 1   | 4,753            | 3,603      | 1,053      | 97         |
> | 2   | 4,635            | 3,418      | 1,069      | 148        |
> | 3   | 4,557            | 3,295      | 1,081      | 181        |
> | 4   | 4,290            | 3,047      | 840        | 339        |
>
> Here, **“Distinct Samples”** refers to the number of unique filtered pairs across all three models for a given $K$. The three rightmost columns indicate how many of these pairs appear in the filtered sets of exactly one, two, or all three models.
>
> Moreover, if the reviewer is interested in how different models perform under identical training data, we would like to point to Figure 2 (the red curve, _without data filtering_). In this setting, we keep the conditions well controlled:
> - All models are trained on exactly the same samples with the same value of $K$.
> - For different values of $K$, the distributions of the current state $S(t)$ and the ground-truth actions are also kept identical across models.
>
> If you have any additional questions about the details, please don't hesitate to reach out. We would be glad to provide further clarification.
>
> ---
>
> >  Question 2: In Table 3, are the task/step instruction baselines also trained with GRPO and the same binary-format reward?
>
> Yes. To ensure a fair comparison, the task-instruction and step-instruction baselines are trained with the same GRPO setup, using the same binary reward and the same hyper-parameter configurations (please refer to Appendix A).
>
> We also matched the training data distribution across tasks: for each sample, the current state $S(t)$ used in the K-step GUI Transition task is identical to the screenshot used in the two instruction-based baselines. This ensures that all models learn from the same ground-truth action distribution.
>
> ---
>
> >  Question 3: If only click points are recorded, how is the box inferred, and how would GUI‑Shift operate on real logs that lack element metadata?
>
> Thank you for raising this question. One reason we use AndroidControl as the training dataset is that it provides screenshots, click points, and VH metadata. To obtain the ground-truth box for a click action from AndroidControl, we identify all GUI elements in the VH whose bounding boxes contain the click point, and then select the element with the smallest area.
>
> **In scenarios where only click points are available and VH metadata is missing**, we outline two practical ways to construct the training signal:
> - **Augment metadata.** Lightweight domain-specific models or current large vision-language models can be used to detect GUI elements and estimate bounding boxes [1] [2]. This approach is relatively inexpensive and can achieve high accuracy in practice.
> - **Design metadata-free rewards.** A reward can be defined solely based on the distance between the predicted and ground-truth click points. For example, a binary reward can be assigned using a distance threshold, or a continuous reward can be used where closer predictions receive higher scores.
>
> These approaches make GUI-Shift applicable even when structured element metadata is not available.
>
> If you have any further questions, please don’t hesitate to let us know. We would be glad to assist in any way we can.
>
> [1] Xie, Mulong, et al. "UIED: a hybrid tool for GUI element detection." _Proceedings of the 28th ACM Joint Meeting on European Software Engineering Conference and Symposium on the Foundations of Software Engineering_. 2020.
>
> [2] Chen, Jieshan, et al. "Object detection for graphical user interface: Old fashioned or deep learning or a combination?." _proceedings of the 28th ACM joint meeting on European Software Engineering Conference and Symposium on the Foundations of Software Engineering_. 2020.
>
> ---
>
> Thank you again for the constructive feedback. We appreciate the opportunity to clarify these points and are very happy to discuss any remaining concerns.

---

### Official Review · Reviewer_QngP · 2025-11-02

**Soundness:** 3
**Presentation:** 3
**Contribution:** 3
**Rating:** 4
**Confidence:** 4

**Summary:**

GUI-Shift is a self-supervised RL framework that applies GRPO to the “K-step GUI Transition” task: given a current screen St and a future screen S_(t+k) taken from real trajectories, the model predicts the first action that initiates the transition. It optimizes with rule-verified rewards (format + action correctness) over a unified 8-type action space, and uses a sampling-and-scoring loop for both training and data filtering. Trained on 2K samples per backbone (k∈{1,2,3,4}) and evaluated on AndroidControl/GUI Odyssey (automation) plus ScreenSpot-v2/Pro (grounding), GUI-Shift reports notable gains—up to +11.2% accuracy on AndroidControl-High and +2.5% on ScreenSpot-v2—without extra alignment.

**Strengths:**

GUI-Shift is practical and scalable because it builds supervision directly from real GUI trajectories (no manual labels) and trains with a GRPO sampling-and-ranking loop that better accommodates multiple plausible actions than single-label SFT, while its rule-verified rewards provide precise, automatic checks on action type and arguments. The unified eight-action interface and “keep only informative cases” filtering keep optimization focused and stable, and dropping explicit reasoning traces cuts training overhead without hurting downstream performance, together yielding consistent gains on mobile GUI automation and even modest improvements on GUI grounding.

**Weaknesses:**

The method treats the trajectory’s first action as the only gold label. Rule-based rewards allow coordinate variance (e.g., any point inside the bbox) but not type/argument alternatives, so equally valid first moves that still reach S_(t+k) get penalized, biasing against strategy-level equivalence.

Training/eval center on AndroidControl and GUI Odyssey (mobile). Grounding sets are included, but not multi-step control. Metrics are mostly single-step TM/EM, no end-to-end success rates, latency/token cost, or cross-OS/desktop multi-app scenarios—so external validity is limited.

Overly literal rewards, weak goal alignment. Rewards favor exact action type/argument matches (e.g., verbatim input_text, specific scroll step) instead of measuring progress toward the target screen S_(t+k). This creates false negatives for synonymous inputs or alternative scrolls and teaches label imitation over goal achievement.

First-step-only training/eval (short-horizon myopia). Given (St, S_(t+k)), the model learns only the first action, and metrics reflect single-step accuracy. This risks poor multi-step planning and error recovery during rollouts, weakening correlation with real agent performance.

**Questions:**

see above

---

> ### Author Response · Authors · 2025-11-20
> **Response to Reviewer QngP**
>
> We appreciate that the reviewer recognizes the practicality and efficiency of our work. We treasure the opportunity to address your concerns.
>
> > Weakness 1 & 3: Rule-based reward design and imitation-style supervision do not fully handle the multi-solution problem in GUI tasks.
>
> Thank you for these thoughtful comments. We agree that our rule-based reward, which emphasizes matches in action types and arguments, does not fully address the multi-solution problem in GUI tasks. This challenge is also present in many other SFT and offline RL settings beyond GUI tasks, where the supervision comes from a single recorded action and alternative valid actions are not encouraged. Even online RL setups encounter similar issues, as progress rewards for GUI operations are difficult to define, while outcome-only rewards tend to be too sparse.
>
> That said, we believe the results in this paper show that GUI-Shift still provides a practical and effective way to reduce annotation costs, and that the same idea can be easily extended to other domains such as web, desktop platforms, or embodied navigation tasks.
>
> We appreciate this insight. Handling multi-solution behavior in GUI environments is an important direction, and we plan to explore progress-based or goal-centered rewards that better reflect whether an action moves toward the target state.
>
> We sincerely appreciate your valuable feedback, and if you have any further questions, we are always happy to assist.
>
> ---
> >  Weakness 2: Evaluation lacks end-to-end success rates and cross-OS or desktop multi-app scenarios.
>
> Thank you for raising this concern. We would first like to clarify that GUI Odyssey [1] already includes tablet interfaces and cross-app trajectories. As shown in Table 5 of Appendix C, Qwen2.5-VL-7B improves by 10.8% on GUI Odyssey, and in Table 6, InternVL3-8B improves by 5.4%. Given that our training data contains only 2K mobile GUI samples, we believe these improvements are reasonable and acceptable. In addition, to further strengthen performance on web or desktop environments, GUI-Shift can be applied to these platforms by constructing the same K-step GUI Transition data from unlabeled trajectories collected from those environments.
>
> For end-to-end performance, please refer to General Response 1. We are actively conducting more experiments on interactive benchmarks with more models.
>
> [1] Lu, Quanfeng, et al. "GUIOdyssey: A Comprehensive Dataset for Cross-App GUI Navigation on Mobile Devices." _Proceedings of the IEEE/CVF International Conference on Computer Vision_. 2025.
>
> ---
> >  Weakness 4: Single-step training and evaluation risk real agent performance.
>
> Thank you for raising this concern. We adopt the first-step prediction in the K-step GUI Transition task training because this setup matches real interaction settings, where each decision is made based on the current screen. In contrast, predicting a full multi-step sequence would require the model to output later actions without seeing the intermediate screens. After the first action, the model would lack access to updated states, making key information such as click locations difficult to infer and difficult to control.
>
> For multi-step evaluation results, please refer to General Response 1.
>
> If you have any other questions or would like further clarification, please feel free to let us know. We would be glad to assist.
>
> ---
> Thank you again for your thoughtful feedback. We appreciate the chance to clarify these points. We hope these clarifications address your concerns, and we are very happy to provide further details if needed.

---

### Official Review · Reviewer_VvL8 · 2025-11-03

**Soundness:** 3
**Presentation:** 3
**Contribution:** 3
**Rating:** 6
**Confidence:** 2

**Summary:**

This paper proposes a self-supervised K-step GUI Transition task, which enables VLMs to learn GUI dynamics by predicting the initial action that causes a transition between two GUI states.

The approach eliminates the need for natural language instructions and enables scalable dataset construction from existing GUI trajectories. Building on this task, the author propose GUI-Shift with reinforcement learning (RL) framework. Experiments show that training on GUI-Shift generalizes well to both GUI automation and grounding tasks, yielding up to an 11.2% increase in GUI automation accuracy.

**Strengths:**

This paper proposed a novel GUI task, k-step GUI transition task, which enables self-supervised training and eliminate dataset construction.

**Weaknesses:**

Despite the efficiency of the method, mechanisms and deeper analysis are somewhat lacking - please see details in questions part.

**Questions:**

1. How to assure the quality of training data? From fig. 1, the task has 2 target, task prediction and action prediction, action prediction can achieve by compared to ground-truth, but how to assure task prediction is aligned with the actual target.
2. Is there an explanation why the shift prediction task could improve the performance of grounding task, which seems to be irrelevant even when k=4.
3. Could GUI-shift generalize to different scenarios and tasks, like out-of-domain tasks.
4. Is there some connections between step K and task difficult or model’s capability.
5. Could model tackle task if the trajectory includes significant state shift across screenshots like mobile application tasks (ac is relative simple)
6. Could K-step GUI Transition task possesses scaling law compared to datasets constructed by general trajectory, further experiment is encourage if calculation resource is enough.

---

> ### Author Response · Authors · 2025-11-20
> **Response to Reviewer VvL8 (Part 1/2)**
>
> We appreciate that the reviewer recognizes the novelty and efficiency of our work. We treasure the opportunity to address your concerns and provide a deeper explanation and analysis of our work.
>
> ---
>
> >  Question 1: How to assure the quality of the training data, and how to assure that task prediction is aligned with the actual target?
>
> Thank you for raising this question. We assure the overall data quality through the **model-specific filtering mechanism** used before GRPO training. As detailed in Section 3.3, we use the K-step GUI Transition data to let the model sample 8 rollouts, and we keep samples where it succeeds in 1-7 of them for the subsequent training. These samples fit the base model’s learning capacity. As discussed in Section 4.3 and Figure 2, this filtering step helps us keep reliable training data and makes the training process more efficient.
>
> We would also like to clarify that the "task prediction" shown in the middle part of Figure 1 is only an intuitive illustration of how the K-step GUI Transition task provides learning signals. In the actual training pipeline, the model only predicts the next GUI action, and action correctness is the sole indicator to confirm that the model has learned the intended transition. No explicit reasoning step is involved in training. To make this clearer, we have updated Figure 1 by changing "Learn from GUI Dynamics" to "Implicit Learning Signal from GUI Dynamics."
>
> If you have any additional questions about the details, please don't hesitate to reach out. We would be glad to provide further clarification.
>
> ---
>
> >  Question 2: Why does the K-step GUI Transition task improve the performance of the grounding task, even when $k=4$?
>
> Thank you for raising this question. This is mainly because the training data includes click actions. To predict a click, the model must locate the correct UI region, which naturally requires grounding. Even when $k=4$, the model still has to find the element in the current state $S(t)$ that can lead toward the target state $S(t+4)$. This process reinforces its ability to locate the correct region and therefore improves grounding performance.
>
> If you have any other questions or need further clarification on the details, please feel free to let us know. We’d be happy to assist you.
>
> ---
>
> >  Question 3: Can GUI-shift generalize to different scenarios and tasks?
>
> Thank you for raising this question. We would like to explain it from three perspectives:
>
> First, based on the current results, the K-step GUI Transition task uses an input format that differs from both textual instruction-based GUI task automation and GUI grounding tasks. Even with this difference, our method still improves performance on both evaluations. This suggests that GUI-Shift is not narrowly fitted to its training setting and shows a degree of robustness and generalization.
>
> Second, from the method itself, GUI-Shift can be applied to other platforms by constructing transition pairs from unlabeled trajectories in web or desktop environments. The same idea can also apply beyond GUI settings. For example, in embodied navigation, K-step visual transitions can serve as learning signals as well.
>
> Third, we additionally evaluate GUI-Shift on an interactive GUI task automation benchmark, which is more challenging and differs from our training data. Under this setting, GUI-Shift-Mimo-RL performs better than its base model. Please refer to General Response 1 for more details on this evaluation.
>
> Overall, we believe that GUI-Shift remains practical across different settings and can transfer beyond mobile GUI tasks.
>
> If anything remains unclear, please feel free to let us know. We would be glad to help.

---

> ### Author Response · Authors · 2025-11-20
> **Response to Reviewer VvL8 (Part 2/2)**
>
> > Question 4: Is there some connections between step K, task difficulty, or model’s capability?
>
> Thank you for raising this question. To study the relationship between the $K$ value, task difficulty, and model capability, we now include a detailed analysis of the reward distributions obtained during the data filtering stage. Please refer to **Figure 4 in Appendix E** for a more direct visualization.
>
> Specifically, our analysis is based on 7,783 shared samples. For fair comparison, we use exactly the same samples for each K across all models, and the current state $S(t)$ and ground-truth actions remain identical across different K values.
>
> We provide detailed analysis from two perspectives:
> - From the perspective of $K$.
> 	- For cases where all 8 rollouts are incorrect, the sample counts follow the pattern $K=4>3>2>1$, indicating that larger $K$ values correspond to higher difficulty.
> 	- For cases where all 8 predictions are correct, the trend reverses to $K=1>2>3>4$, showing that smaller $K$ values are easier for the models.
> - From the perspective of model capability.
> 	- As shown in Table 1 and Table 2, Mimo-VL-7B-SFT and Mimo-VL-7B-RL perform better than InternVL3-8B on four GUI-related benchmarks. We may reasonably consider the Mimo models to have stronger GUI capability. A similar pattern appears during data filtering: both Mimo models have noticeably more samples with all correct predictions than InternVL3-8B, suggesting that models with stronger GUI capability perform better on the K-step GUI Transition task.
>
> Overall, these findings reflects that the K-step GUI Transition task becomes more challenging as $K$ increases, and that models with stronger GUI capability achieve better performance on this task, which aligns with intuition.
>
> We appreciate this question. It encourages us to examine the links among $K$, task difficulty, and model ability more carefully. If you have any other questions or need further clarification on the details, please feel free to let us know. We’d be happy to assist you.
>
> ---
>
> >  Question 5: Can GUI-Shift handle trajectories that involve large state shifts?
>
> Thank you for the question. In addition to AndroidControl [1], we also evaluate GUI-Shift on GUI Odyssey [2], which contains both phone and tablet interfaces and includes cross-app trajectories. This setting is more challenging, with larger state shifts between consecutive screens compared with AndroidControl.
>
> As shown in Table 5 in the Appendix C, without data filtering, GUI-Shift-Qwen improves Exact Match on GUI Odyssey by 8.2-10.8%, and GUI-Shift-InternVL improves by 3.0%. With data filtering, as shown in Table 6 in the Appendix C, GUI-Shift-InternVL improves by up to 5.4%. These results indicate that GUI-Shift can handle trajectories with larger variation and remains effective under such conditions.
>
> We also evaluate GUI-Shift on AndroidWorld [3], an interactive GUI automation benchmark that is more challenging and involves large state shifts. We observe that GUI-Shift-Mimo-RL performs better than its base model. For more details, please refer to General Response 1.
>
> We welcome any further questions and are happy to offer more explanation where needed.
>
> [1] Li, Wei, et al. "On the effects of data scale on ui control agents." _Advances in Neural Information Processing Systems_ 37 (2024): 92130-92154.
>
> [2] Lu, Quanfeng, et al. "GUIOdyssey: A Comprehensive Dataset for Cross-App GUI Navigation on Mobile Devices." _Proceedings of the IEEE/CVF International Conference on Computer Vision_. 2025.
>
> [3] Rawles, Christopher, et al. "Androidworld: A dynamic benchmarking environment for autonomous agents." _arXiv preprint arXiv: 2405.14573_ (2024).
>
> ---
>
> >  Question 6: Can K-step GUI Transition task possesses scaling law compared to datasets constructed by general trajectories, further experiment is encourage if calculation resource is enough.
>
> Thank you for this suggestion. Within the limited rebuttal period, we did our best to conduct additional experiments on InternVL3-8B by scaling its training data from 2K to 6K, following the same filtering and training setup described in the original paper.
>
> Specifically, relative to the 2K setting, GUI-Shift-Intern shows an additional gain of 1.7% when trained with 6K samples on AndroidControl-Low, and a further 2.0% improvement when trained with 3K samples on AndroidControl-High. In general, we observe an overall upward trend in accuracy as the data size increases, with only small fluctuations across different scales.  Please refer to **Figure 5 in Appendix F** for more details.
>
> If you have any other questions or need further clarification on the details, please feel free to let us know. We’d be happy to assist you.
>
> ---
>
> Thank you again for the constructive feedback. We hope these clarifications address your concerns, and we are very happy to provide further details if needed.

---

> > ### Comment · Reviewer_VvL8 · 2025-11-26
> >
> > Thank you for the detailed clarifications and for the additional experiments.  Based on the feedback (as well as the other reviewers' comments), I am happy to maintain my positive score, and I have increased my confidence about it.

---

> > > ### Author Response · Authors · 2025-11-26
> > >
> > > We sincerely appreciate the reviewer’s positive feedback and the increased confidence in the evaluation! Thank you for taking the time to read our additional clarifications and experiments. Your support and recognition mean a lot to us, and we are truly grateful for the constructive comments throughout the discussion.

---

### Public Comment · ~Shuquan_Lian2 · 2025-11-13

Dear Authors,

I enjoyed reading your paper and found the results on Android Control and GUI-Odyssey particularly impressive. It is great to see progress in this direction.

I am writing to bring to your attention a relevant recent work, UI-AGILE: Advancing GUI Agents with Effective Reinforcement Learning and Precise Inference-Time Grounding, which evaluates on Android Control, ScreenSpot-v2, ScreenSpot-Pro. This work also leverages RL to train GUI agents, employing a cropping mechanism to alleviate the challenges of GUI grounding during both training and inference.

Interestingly, looking at the results, we observe a performance trade-off: while UI-AGILE achieve higher performance on ScreenSpot-Pro, your proposed approach demonstrates superior performance on Android Control with less training samples (9K vs. 2K).

Including this comparison could be valuable for the readers to understand the strengths of different methodologies across different settings.

Best regards

---

### Author Response · Authors · 2025-11-20
**General Response to Reviewers (Part 1/2)**

We thank all the reviewers for the effort engaged in the review phase and the constructive feedback on our submission. We are delighted that our framework is described as *"This paper proposed a novel GUI task"* (reviewer VvL8), *"GUI-Shift is practical and scalable"* (reviewer QngP), *"The K-step Transition objective is a compact way to leverage unlabeled trajectories at scale"* (reviewer 7wjQ), and *" This addresses the scalability bottleneck in GUI agent training data. The experimental quality is a major strength. The paper is exceptionally clear."* (reviewer hoLT).

GUI-Shift takes the future state $S(t+k)$ as the visual instruction to guide model optimization, which eliminates the need for high-cost textual annotations and avoids the extra noise introduced by annotation quality. We validate GUI-Shift on four VLMs and four GUI-related benchmarks. We also conduct extensive experiments to examine the effectiveness of each design choice in this framework, including the choice of K, the model-specific data filtering mechanism, task formulation, reasoning configurations, and training algorithms. Although we focus on the mobile platform in this paper, we believe that GUI-Shift can be extended to both web and desktop platforms with minimal effort.

We treasure the opportunity to discuss our work with all reviewers. In the following, we provide detailed responses to address the concerns and suggestions raised by each reviewer.

**We have revised our paper (highlighted in blue text color). The modifications are summarized as follows.**

1. (For Reviewer VvL8) We update **Figure 1** by changing "Learn from GUI Dynamics" to "Implicit Learning Signal from GUI Dynamics" to better clarify that this part represents an implicit learning signal rather than an explicit reasoning step.
2. (For Reviewer VvL8, 7wjQ) We add a detailed analysis of the reward distributions obtained during the data filtering stage in **Appendix E**. In short, we observe that the K-step GUI Transition task becomes harder as $K$ increases, and models with stronger GUI skills achieve higher rewards.
3. (For Reviewer VvL8) We conduct additional experiments on InternVL3-8B by scaling its training data from 2K to 6K. The corresponding figures are now included in **Appendix F**. In general, we observe an overall upward trend in accuracy as the data size increases.

---

> ### Author Response · Authors · 2025-11-20
> **General Response to Reviewers (Part 2/2)**
>
> **Regarding the reviewers' common comments, we provide the following general response.**
>
> > General Response 1: End-to-end success rate of GUI-Shift on AndroidControl and AndroidWorld [1].
>
> To address this concern, we report end-to-end success rates on AndroidControl, where an episode is considered successful only when **all** steps are correct. If any step is incorrect, the entire trajectory is counted as a failure. We also extend our evaluation to AndroidWorld, an interactive GUI task automation benchmark with 116 tasks.
>
> ---
> **For AndroidControl**, the following tables report the success rates for all four GUI-Shift variants (AC-Low: AndroidControl-Low; AC-High: AndroidControl-High):
>
> | Model                | AC-Low | AC-High | Model                  | AC-Low | AC-High |
> | -------------------- | ------ | ------- | ---------------------- | ------ | ------- |
> | Qwen2.5-VL-7B        | 50.2   | 22.4    | InternVL3-8B           | 68.3   | 15.4    |
> | GUI-Shift-Qwen (K=1) | 67.5   | 29.9    | GUI-Shift-Intern (K=1) | 60.7   | 18.9    |
> | GUI-Shift-Qwen (K=2) | 65.1   | 29.9    | GUI-Shift-Intern (K=2) | 60.3   | 19.2    |
> | GUI-Shift-Qwen (K=3) | 69.3   | 29.4    | GUI-Shift-Intern (K=3) | 61.0   | 19.2    |
> | GUI-Shift-Qwen (K=4) | 67.6   | 28.7    | GUI-Shift-Intern (K=4) | 61.1   | 21.9    |
>
> | Model                    | AC-Low | AC-High | Model                   | AC-Low | AC-High |
> | ------------------------ | ------ | ------- | ----------------------- | ------ | ------- |
> | Mimo-VL-7B-SFT           | 48.4   | 16.4    | Mimo-VL-7B-RL           | 53.3   | 18.0    |
> | GUI-Shift-Mimo-SFT (K=1) | 72.4   | 32.1    | GUI-Shift-Mimo-RL (K=1) | 76.3   | 32.2    |
> | GUI-Shift-Mimo-SFT (K=2) | 73.9   | 33.1    | GUI-Shift-Mimo-RL (K=2) | 71.8   | 34.6    |
> | GUI-Shift-Mimo-SFT (K=3) | 75.7   | 34.1    | GUI-Shift-Mimo-RL (K=3) | 71.0   | 33.6    |
> | GUI-Shift-Mimo-SFT (K=4) | 74.2   | 31.1    | GUI-Shift-Mimo-RL (K=4) | 69.5   | 33.7    |
>
> As shown above, GUI-Shift brings clear improvements in multi-step scenarios. For example, on AndroidControl-Low, Mimo-VL-7B-SFT improves from 48.4% to 75.7%, and on AndroidControl-High, it improves from 16.4% to 34.1%.
>
> ---
> **For AndroidWorld**, we evaluate Mimo-VL-7B-RL and GUI-Shift-Mimo-RL (K=1) under the M3A agent setting. The results for pass@1, pass@3, and pass@5 are as follows:
>
> | Model                   | Pass@1         | Pass@3         | Pass@5         |
> | ----------------------- | -------------- | -------------- | -------------- |
> | Mimo-VL-7B-RL           | 16/116 (13.8%) | 22/116 (19.0%) | 23/116 (19.8%) |
> | GUI-Shift-Mimo-RL (K=1) | 17/116 (14.7%) | 23/116 (19.8%) | 27/116 (23.3%) |
>
> With all settings the same, GUI-Shift-Mimo-RL performs better than its base model. In particular, at pass@5, it improves from 19.8% to 23.3%.
>
> Taken together, these results indicate that GUI-Shift strengthens multi-step performance in a practical and efficient way. We are actively expanding our evaluation to more models and interactive benchmarks, and we will share new results as they become available.
>
> [1] Rawles, Christopher, et al. "Androidworld: A dynamic benchmarking environment for autonomous agents." _arXiv preprint arXiv:2405.14573_ (2024).
>
> ---
> Once again, we thank all reviewers for their invaluable suggestions and insights, which have greatly contributed to improving our submission. Please feel free to let us know if any further clarification would be helpful for the final assessment.

---

> ### Author Response · Authors · 2025-11-26
> **General Response to Reviewers (Updated Results on AndroidWorld)**
>
> We sincerely thank all the reviewers for their time and constructive feedback on our paper. We deeply value each reviewer's insightful comments and have been actively conducting additional experiments to address your concerns and strengthen our work.
>
> In our first response, following the suggestions from reviewers **QngP** and **7wjQ** regarding end-to-end evaluation, we reported **the end-to-end success rates on AndroidControl** and **the Pass@k results on AndroidWorld**, an interactive benchmark with 116 challenging tasks. We observed substantial improvements in end-to-end success rates on AndroidControl, and GUI-Shift-Mimo-RL (k=1) achieved a 3.5% improvement over the baseline at Pass@5 on AndroidWorld.
>
> We are actively continuing to evaluate additional models. We now provide updated results including GUI-Shift-Mimo-RL (k=4), as well as Pass@1 and Pass@3 results for Mimo-VL-7B-SFT and GUI-Shift-Mimo-SFT (k=1, k=2). All results on AndroidWorld are evaluated under the M3A agent setting. The current AndroidWorld results are shown below:
>
> | Model                   | Pass@1         | Pass@3         | Pass@5         |
> | ----------------------- | -------------- | -------------- | -------------- |
> | Mimo-VL-7B-RL           | 16/116 (13.8%) | 22/116 (19.0%) | 23/116 (19.8%) |
> | GUI-Shift-Mimo-RL (k=1) | 17/116 (14.7%) | 23/116 (19.8%) | 27/116 (23.3%) |
> | GUI-Shift-Mimo-RL (k=4) | 18/116 (15.5%) | 23/116 (19.8%) | 26/116 (22.4%) |
>
> | Model                    | Pass@1         | Pass@3         |
> | ------------------------ | -------------- | -------------- |
> | Mimo-VL-7B-SFT           | 7/116 (6.0%)   | 17/116 (14.7%) |
> | GUI-Shift-Mimo-SFT (k=1) | 18/116 (15.5%) | 22/116 (19.0%) |
> | GUI-Shift-Mimo-SFT (k=2) | 14/116 (12.1%) | 17/116 (14.7%) |
>
> These results demonstrate that GUI-Shift-Mimo-RL (k=1, k=4) both improve upon Mimo-VL-7B-RL, particularly at Pass@5 with gains of 3.5% and 2.6% respectively.
>
> GUI-Shift-Mimo-SFT (k=1) achieves improvements over Mimo-VL-7B-SFT: **a 2.6x improvement at Pass@1 (from 6.0% to 15.5%)** and **a 4.3% gain at Pass@3 (from 14.7% to 19.0%)**. GUI-Shift-Mimo-SFT (k=2) also shows **a 2x improvement at Pass@1 (from 6.0% to 12.1%)**. These findings indicate that GUI-Shift, as a self-supervised approach, effectively improves not only static single-step task accuracy (as shown in Tables 1 and 2 in our paper) but also dynamic end-to-end task performance.
>
> We are continuing to evaluate additional models and will update any new results here. These experimental findings will be incorporated into the revised paper.
>
> We sincerely thank all reviewers again for the valuable feedback and hope that these additional experiments help address your concerns. If you have any further questions or suggestions, please feel free to share them. We would be more than happy to provide additional clarification.

---

> ### Author Response · Authors · 2025-11-28
> **General Response to Reviewers (Update on AndroidWorld Results)**
>
> Dear Reviewers,
>
> We would like to provide a brief update on our ongoing evaluation. We now have Pass@5 results for Mimo-VL-7B-SFT and two GUI-Shift-Mimo-SFT models on AndroidWorld:
>
> |Model|Pass@1|Pass@3|Pass@5|
> |---|---|---|---|
> |Mimo-VL-7B-SFT|7/116 (6.0%)|17/116 (14.7%)|21/116 (18.1%)|
> |GUI-Shift-Mimo-SFT (k=1)|18/116 (15.5%)|22/116 (19.0%)|28/116 (24.1%)|
> |GUI-Shift-Mimo-SFT (k=2)|14/116 (12.1%)|17/116 (14.7%)|25/116 (21.6%)|
>
> We observe that GUI-Shift-Mimo-SFT (k=1) achieves a **6.0% improvement** over the baseline at Pass@5 (from 18.1% to 24.1%), and GUI-Shift-Mimo-SFT (k=2) demonstrates a **3.5% gain** (from 18.1% to 21.6%). These results indicate that GUI-Shift helps multi-step interactive tasks reliably.
>
> We are actively running more evaluations, and additional results for other models will be reported as soon as they become available.
>
> If you have any concerns or questions regarding these results or any other aspects of our work, please do not hesitate to let us know. We are happy to provide further clarification or conduct additional experiments as needed.
>
> Thank you again for your valuable feedback and continued engagement with our work.

---

### Author Response · Authors · 2025-11-29
**Summary of Responses to Reviewers (Part 1/3)**

We thank all the reviewers for the effort engaged in the review phase and the constructive feedback on our submission. We are delighted that our framework is described as *"This paper proposed a novel GUI task"* (reviewer VvL 8), *"GUI-Shift is practical and scalable"* (reviewer QngP), *"The K-step Transition objective is a compact way to leverage unlabeled trajectories at scale"* (reviewer 7 wjQ), and *"This addresses the scalability bottleneck in GUI agent training data. The experimental quality is a major strength. The paper is exceptionally clear."* (reviewer hoLT).

GUI-Shift takes the future state $S(t+k)$ as the visual instruction to guide model optimization, which eliminates the need for high-cost textual annotations and avoids the extra noise introduced by annotation quality. We validate GUI-Shift on four VLMs and four GUI-related benchmarks. We also conduct extensive experiments to examine the effectiveness of each design choice in this framework, including the choice of K, the model-specific data filtering mechanism, task formulation, reasoning configurations, and training algorithms. Although we focus on the mobile platform in this paper, we believe that GUI-Shift can be extended to both web and desktop platforms with minimal effort.

We sincerely appreciate the opportunity to discuss our work with all reviewers. During the rebuttal period, we have actively conducted additional experiments and provided deeper analysis to address each reviewer's questions and concerns.

For clarity and convenience, we summarize our responses to each reviewer below.

---

**For Reviewer VvL8**

- **Question 1: Alignment of learning objectives.** We refined Figure 1 and updated the paper.
- **Question 2: How K-step GUI Transition benefits downstream GUI grounding.** We provided a deeper explanation.
- **Question 3: Generalization beyond mobile GUI scenarios.** We provided feasible solutions.
- **Question 4: Relationship between K value, task difficulty, and model capability of the K-step GUI Transition task.** We additionally analyzed the reward distributions obtained during the data filtering stage and updated the visualization results in Figure 4 (Appendix E).
- **Question 5: Performance on more difficult tasks.** We additionally evaluated trained GUI-Shift models on a more challenging and dynamic benchmark, AndroidWorld [1].
- **Question 6: Performance when data size scales.** We additionally scaled training data from 2K to 6K and updated the detailed results in Figure 5 (Appendix F).

We thank the reviewer for taking the time to read our additional clarifications and experiments, and for improving the confidence score from 2 to 4.

[1] Rawles, Christopher, et al. "Androidworld: A dynamic benchmarking environment for autonomous agents." _arXiv preprint arXiv: 2405.14573_ (2024).

---

**For Reviewer QngP**

- **Weakness 1 & 3: Concern about handling the multi-solution problem in GUI tasks.** We first explained the rationale and the advantage of imitation-style learning in most GUI scenarios. We then conducted additional end-to-end evaluations on both AndroidControl and the more challenging, interactive AndroidWorld benchmark. Results demonstrate that GUI-Shift not only improves single-step accuracy but also enhances multi-step success rates in end-to-end settings, indicating that training gains at the action level can also benefit tasks with multiple valid solution paths.
- **Weakness 2: End-to-end evaluation and cross-platform evaluation.** We reported end-to-end success rates on AndroidControl and conducted additional evaluations on AndroidWorld. We clarified that GUI Odyssey, as shown in the original paper, already includes cross-platform and cross-app tasks. While GUI-Shift focuses on mobile GUI scenarios with 2K mobile samples, we provided feasible solutions for extending to other platforms.
- **Weakness 4: Single-step training and evaluation.**  We provided end-to-end, multi-step evaluation results on both AndroidControl and AndroidWorld. These results validate the effectiveness of GUI-Shift in real-world, sequential task scenarios.

---

> ### Author Response · Authors · 2025-11-29
> **Summary of Responses to Reviewers (Part 2/3)**
>
> **For Reviewer 7wjQ**
>
> - **Weakness 1: Training data source selection.** We explained the reason why we choose AndroidControl and clarified that we do not use annotations in GUI-Shift training.
> - **Weakness 2: Performance on GUI Odyssey and ScreenSpot-Pro.** We provided analysis of the performance on these two benchmarks and clarified that the GUI-Shift training paradigm could be transferred to web or desktop platforms to further boost performance beyond mobile platforms.
> - **Weakness 3: End-to-end performance.** We provided end-to-end results on AndroidControl and conducted additional evaluation on AndroidWorld.
> - **Question 1: Filtering data statistics.** We provided detailed statistics on the valid data percentage for each model and each K, and quantified the inter-model overlap of filtered pairs.
> - **Question 2: Baseline training setting.** We clarified the training settings for the task/step instruction baselines.
> - **Question 3: Apply to datasets without element metadata.** We provided two feasible solutions on how to apply GUI-Shift to training datasets without element metadata.
>
> ---
>
> **For Reviewer hoLT**
>
> - **Question 2: Discussion on predicting the full sequence rather than single next step.** We clarified why we adopt predicting the next action that initiates the GUI state transition and discussed the feasibility of predicting multiple consecutive actions.
> - **Question 3: The generality of model-specific data filtering mechanism.** We provided the data filtering results for Qwen2.5-VL-7B to show why we do not apply data filtering on this model. We further provided feasible solutions on how to apply this data filtering mechanism to other models with poor performance on the K-step GUI Transition task.
> - **Question 4: Discussion on omitting reasoning traces during training.** We explained our experimental observations and provided explanations for omitting reasoning traces.
> - **Question 5: Discussion on the optimal value of K.** We provided further clarification and analysis of how K affects training performance.
>
> ---
> We sincerely thank all reviewers again for their valuable feedback and for helping us improve our work.

---

> ### Author Response · Authors · 2025-11-29
> **Summary of Responses to Reviewers (Part 3/3: End-to-End Performance on AndroidControl and AndroidWorld)**
>
> For convenient review, we summarize the end-to-end performance on AndroidControl and AndroidWorld below.
>
> **Evaluation Setup:**
> - **AndroidControl:** We report end-to-end success rates where an episode is considered successful only when all steps are correct. If any step is incorrect, the entire trajectory is counted as a failure.
> - **AndroidWorld:** We evaluate on this interactive GUI task automation benchmark with 116 tasks under the M3A agent setting.
>
> ---
> **AndroidControl Results:**
>
> The following tables report success rates for all four GUI-Shift variants (AC-Low: AndroidControl-Low; AC-High: AndroidControl-High):
>
> | Model                | AC-Low | AC-High | Model                  | AC-Low | AC-High |
> | -------------------- | ------ | ------- | ---------------------- | ------ | ------- |
> | Qwen2.5-VL-7B        | 50.2   | 22.4    | InternVL3-8B           | 68.3   | 15.4    |
> | GUI-Shift-Qwen (K=1) | 67.5   | 29.9    | GUI-Shift-Intern (K=1) | 60.7   | 18.9    |
> | GUI-Shift-Qwen (K=2) | 65.1   | 29.9    | GUI-Shift-Intern (K=2) | 60.3   | 19.2    |
> | GUI-Shift-Qwen (K=3) | 69.3   | 29.4    | GUI-Shift-Intern (K=3) | 61.0   | 19.2    |
> | GUI-Shift-Qwen (K=4) | 67.6   | 28.7    | GUI-Shift-Intern (K=4) | 61.1   | 21.9    |
>
> | Model                    | AC-Low | AC-High | Model                   | AC-Low | AC-High |
> | ------------------------ | ------ | ------- | ----------------------- | ------ | ------- |
> | Mimo-VL-7B-SFT           | 48.4   | 16.4    | Mimo-VL-7B-RL           | 53.3   | 18.0    |
> | GUI-Shift-Mimo-SFT (K=1) | 72.4   | 32.1    | GUI-Shift-Mimo-RL (K=1) | 76.3   | 32.2    |
> | GUI-Shift-Mimo-SFT (K=2) | 73.9   | 33.1    | GUI-Shift-Mimo-RL (K=2) | 71.8   | 34.6    |
> | GUI-Shift-Mimo-SFT (K=3) | 75.7   | 34.1    | GUI-Shift-Mimo-RL (K=3) | 71.0   | 33.6    |
> | GUI-Shift-Mimo-SFT (K=4) | 74.2   | 31.1    | GUI-Shift-Mimo-RL (K=4) | 69.5   | 33.7    |
>
> As shown above, GUI-Shift brings clear improvements in multi-step scenarios. For example, on AndroidControl-Low, Mimo-VL-7B-SFT improves from 48.4% to 75.7%, and on AndroidControl-High, it improves from 16.4% to 34.1%.
>
> ---
>
> **AndroidWorld Results:**
>
> For AndroidWorld, we evaluate Mimo-VL-7B-RL, GUI-Shift-Mimo-RL, Mimo-VL-7B-SFT, GUI-Shift-Mimo-SFT under the M3A agent setting. The results for pass@1 , pass@3 , and pass@5 are as follows:
>
> | Model                   | Pass@1         | Pass@3         | Pass@5         |
> | ----------------------- | -------------- | -------------- | -------------- |
> | Mimo-VL-7B-RL           | 16/116 (13.8%) | 22/116 (19.0%) | 23/116 (19.8%) |
> | GUI-Shift-Mimo-RL (K=1) | 17/116 (14.7%) | 23/116 (19.8%) | 27/116 (23.3%) |
> | GUI-Shift-Mimo-RL (K=4) | 18/116 (15.5%) | 23/116 (19.8%) | 26/116 (22.4%) |
>
> | Model                    | Pass@1         | Pass@3         | Pass@5         |
> | ------------------------ | -------------- | -------------- | -------------- |
> | Mimo-VL-7B-SFT           | 7/116 (6.0%)   | 17/116 (14.7%) | 21/116 (18.1%) |
> | GUI-Shift-Mimo-SFT (K=1) | 18/116 (15.5%) | 22/116 (19.0%) | 28/116 (24.1%) |
> | GUI-Shift-Mimo-SFT (K=2) | 14/116 (12.1%) | 17/116 (14.7%) | 25/116 (21.6%) |
> | GUI-Shift-Mimo-SFT (K=3) | 19/116 (16.4%) | 24/116 (20.7%) | 25/116 (21.6%) |
> | GUI-Shift-Mimo-SFT (K=4) | 15/116 (12.9%) | 23/116 (19.8%) | 30/116 (25.9%) |
>
> **Key results:**
> - GUI-Shift-Mimo-RL (K=1, K=4) improve upon Mimo-VL-7B-RL, with gains of 3.5% and 2.6% at Pass@5.
> - GUI-Shift-Mimo-SFT (K=1) achieves clear improvements over Mimo-VL-7B-SFT: **a 2.6x improvement at Pass@1 (from 6.0% to 15.5%), a 4.3% gain at Pass@3 (from 14.7% to 19.0%) and a 6.0% improvement at Pass@5 (from 18.1% to 24.1%)**.
> - GUI-Shift-Mimo-SFT (K=2) shows **a 2x improvement at Pass@1 (from 6.0% to 12.1%)** and **a 3.5% improvement at Pass@5 (from 18.1% to 21.6%)**.
> - GUI-Shift-Mimo-SFT (K=3) also shows **a 2.7x improvement at Pass@1 (from 6.0% to 16.4%)**, **a 6.0% improvement at Pass@3 (from 14.7% to 20.7%)**, and **a 3.5% improvement at Pass@5 (from 18.1% to 21.6%)**.
> - GUI-Shift-Mimo-SFT (K=4) achieves **a 7.8% improvement at Pass@5 (from 18.1% to 25.9%)**.
>
> These results demonstrate that GUI-Shift reliably improves performance on multi-step interactive tasks.
>
> ---
>
> We are actively running more evaluations, and additional results for other models will be updated here as soon as they become available.
>
> Once again, we thank all reviewers for their valuable feedback and engagement with our work.

---

### Meta-Review · Area_Chair_o36r · 2026-01-04

**Summary:**

This paper proposes a self-supervised reinforcement learning framework, termed **GUI-Shift**, for training vision–language–model-based GUI agents by leveraging unlabeled GUI trajectories through a novel K-step GUI Transition task. The core idea is to predict the initial action that induces a transition between two GUI states, thereby avoiding reliance on costly and noisy natural language annotations. Extensive experiments across multiple VLM backbones and benchmarks demonstrate consistent performance gains on both GUI automation and grounding tasks.

In general, the paper is well-motivated and technically sound, addressing an important scalability bottleneck in GUI agent training. Its main strengths lie in the simplicity and generality of the proposed self-supervised objective, the practical RL formulation with verifiable rewards, and the thorough experimental validation across diverse settings, which together provide convincing evidence of the method’s effectiveness.

However, there are some concerns about the paper, as pointed out by the reviewers, particularly regarding the handling of multiple valid action sequences, the reliance on mobile GUI datasets for training and evaluation, and the initial focus on single-step prediction rather than full end-to-end task success. Reviewers also noted that the reward design may bias the model toward imitation of specific actions instead of more goal-oriented behavior, and that cross-platform generalization, while promising, is only partially demonstrated.

During the rebuttal, the authors tried to address those concerns, especially about end-to-end performance and generalization, by providing additional experiments on interactive benchmarks, deeper analysis of the K-step parameter and data filtering mechanism, and clarifications on why the design choices remain appropriate and effective in practice.

**Reviewer Concerns:**

Reviewer VvL8

Addressed: Objective alignment, effect of K-step transition on grounding, relationship between K and task difficulty, generalization, and added end-to-end and scaling experiments.
Outstanding: No significant.

Reviewer QngP

Addressed: End-to-end multi-step evaluation, clarification of cross-app settings, and empirical evidence that action-level gains improve rollout success.
Outstanding: Multi-solution reward design and limited cross-OS/desktop evaluation.

Reviewer 7wjQ

Addressed: Data source choice, filtering statistics and overlap, baseline fairness, metadata-free applicability, and end-to-end AndroidWorld results.
Outstanding: Broader non-mobile empirical validation.

Reviewer hoLT

Addressed: First-action vs. sequence prediction, data filtering generality, omission of reasoning traces, and analysis of K.
Outstanding: Deeper theoretical understanding and broader validation.

**Reviewer Scores:**

Reviewer VvL8
Maintains a positive score

Reviewer QngP
Maintains a positive score

Reviewer 7wjQ
Likely score change: Maintains or slightly increases to marginally above acceptance.

Reviewer hoLT
Maintains a positive score

---

### Decision · Program_Chairs · 2026-01-26

Accept (Poster)